# Mitochondrial adaptor TRAK2 activates and functionally links opposing kinesin and dynein motors

Adam R. Fenton [1,2,3,4], Thomas A. Jongens [1,3,5✉] & Erika L. F. Holzbaur [2,3,4,5✉]

Mitochondria are transported along microtubules by opposing kinesin and dynein motors. Kinesin-1 and dynein-dynactin are linked to mitochondria by TRAK proteins, but it is unclear how TRAKs coordinate these motors. We used single-molecule imaging of cell lysates to show that TRAK2 robustly activates kinesin-1 for transport toward the microtubule plus-end. TRAK2 is also a novel dynein activating adaptor that utilizes a conserved coiled-coil motif to interact with dynein to promote motility toward the microtubule minus-end. However, dynein-mediated TRAK2 transport is minimal unless the dynein-binding protein LIS1 is present at a sufficient level. Using co-immunoprecipitation and co-localization experiments, we demonstrate that TRAK2 forms a complex containing both kinesin-1 and dynein-dynactin. These motors are functionally linked by TRAK2 as knockdown of either kinesin-1 or dynein-dynactin reduces the initiation of TRAK2 transport toward either microtubule end. We propose that TRAK2 coordinates kinesin-1 and dynein-dynactin as an interdependent motor complex, providing integrated control of opposing motors for the proper transport of mitochondria.

[1] Department of Genetics, University of Pennsylvania Perelman School of Medicine, Philadelphia, PA, USA. [2] Department of Physiology, University of Pennsylvania Perelman School of Medicine, Philadelphia, PA, USA. [3] Cell and Molecular Biology Graduate Group, University of Pennsylvania Perelman School of Medicine, Philadelphia, PA, USA. [4] Pennsylvania Muscle Institute, University of Pennsylvania Perelman School of Medicine, Philadelphia, PA, USA. [5]These authors contributed equally: Thomas A. Jongens, Erika L. F. Holzbaur. ✉email: jongens@pennmedicine.upenn.edu; holzbaur@pennmedicine.upenn.edu

Microtubule motors drive the transport of many organelles within the cell. Cargoes are transported to the microtubule plus-end by kinesin motors or to the microtubule minus-end by cytoplasmic dynein 1 (dynein). Whereas individual motors move unidirectionally along microtubules, many cellular cargoes move bidirectionally. For these cargoes, the activities of bound kinesin and dynein motors must be precisely coordinated to ensure proper transport and localization. Despite progress in understanding motor regulation, the mechanisms coordinating opposing motors for organelle transport remain unclear[1].

Mitochondria are actively shuttled along the microtubule cytoskeleton to meet local energy needs. This transport is critical in highly extended cells, such as neurons, whose axons can grow to lengths on the meter scale. Within neurons, mitochondria undergo long-range transport to meet local energy demands and maintain neuronal homeostasis[2]. Loss of mitochondrial transport results in defective neurotransmission and neurodegeneration, highlighting the importance of proper mitochondrial transport for neuronal function[3–7]. Consistently, defective mitochondrial transport and function are implicated in the pathogenesis of neurological diseases, such as Alzheimer's disease[8,9], Parkinson's disease[10,11], and amyotrophic lateral sclerosis[12]. In cells, mitochondria exhibit bidirectional motility, in which transport to either the microtubule plus- or minus-end is punctuated by directional switching or periods of stationary docking. Mitochondrial transport to the microtubule plus-end is mediated primarily by the kinesin-1 (KIF5) family of motors whereas transport to the minus-end is mediated by dynein and its partner complex, dynactin[13,14]. These motor proteins are linked to mitochondria by a conserved complex of motor-adaptor proteins. The TRAK/Milton family of proteins act as mitochondrial motor adaptors that connect kinesin-1 and dynein–dynactin to the mitochondrial outer membrane protein Miro[14–17]. This motor-adaptor function is conserved from *Drosophila* Milton to its mammalian orthologs, TRAK1 and TRAK2. Both TRAK and Miro motor-adaptor proteins are essential for proper mitochondrial distribution and transport in neurons[3,5–7,14].

Despite the essential role for TRAK proteins in mitochondrial transport, little is known about the molecular basis by which TRAKs interact with microtubule motors. Mass spectrometry and co-immunoprecipitation experiments indicate that TRAK1 and TRAK2 interact with kinesin-1 and dynein–dynactin[14,18–21]. However, endogenous KIF5B shows higher binding to TRAK1 than TRAK2, suggesting that TRAK2 has a weaker interaction with kinesin-1[14]. These TRAK-specific interactions with kinesin-1 are thought to account for differences in the localization and function of TRAKs in the axon and dendrites of neurons[14]. Overexpressed TRAK1 promotes plus-end-directed mitochondrial transport whereas overexpressed TRAK2 promotes minus-end-directed mitochondrial transport in mouse embryonic fibroblasts[22]. As a result, current models have proposed that TRAK2 preferentially promotes dynein-mediated transport toward the microtubule minus-end[14,22].

Activation of mammalian dynein requires dynactin and an activating adaptor[23,24]. The formation of a dynein–dynactin-adaptor complex aligns the dynein motor domains for processive transport[25]. Activating adaptors enhance the stability of the dynein–dynactin complex and allow for cargo-specific recruitment of dynein, enabling cargo transport toward the microtubule minus-end[26,27]. Although activating adaptors can vary greatly in structure and function, many activating adaptors have similar interactions with dynein and dynactin. Structural work on dynein–dynactin in complex with BICD2, BICDR1, or HOOK3 showed that each activating adaptor contains an extended coiled-coil domain that binds along the length of

dynactin's 37 nm Arp1 filament[28,29]. All verified activating adaptors contain an extended coiled-coil domain that is sufficient to span this distance[27]. Many of these adaptors have conserved features flanking this coiled-coil domain: a coiled-coil 1 box (CC1-Box) at the N-terminus and a Spindly motif at the C-terminus. Structural studies on BICD2 and Spindly indicate that the CC1-Box binds to dynein light intermediate chain (LIC1) and the Spindly motif binds to the dynactin pointed-end complex[30,31]. Both the CC1-Box and Spindly motif are conserved in TRAKs, where they flank a ~300 amino acid region predicted to form two coiled-coil domains (Fig. 1a). The conservation of these elements within TRAKs suggests that this coiled-coil region scaffolds the dynein–dynactin complex and thus activates dynein. However, the proposed role of TRAK proteins as dynein activating adaptors has yet to be experimentally validated.

The mechanism by which TRAKs coordinate opposing kinesin and dynein motors for mitochondrial transport has remained a perplexing topic. Previous studies have mapped the binding of kinesin-1 to the N-terminal coiled-coil region of TRAK2 (aa 124–283)[14,18]. This region lies within the predicted dynein–dynactin interface of TRAK2 (Fig. 1a). The overlap of the kinesin-1 and dynein–dynactin interfaces on TRAK2 raises the question of whether these opposing motors can simultaneously bind to TRAK2. Further, investigations of mitochondrial transport in neurons demonstrated that knockdown or inhibition of kinesin-1, dynein, or dynactin individually was sufficient to inhibit mitochondrial motility in both directions, suggesting that the activity of one motor is required for the activity of the other[13,14,32–34]. This paradox of motor co-dependence has been a major challenge for understanding the control of bidirectional mitochondrial transport, as the activities of these motors are difficult to uncouple[1].

In light of these observations, we sought to develop a system to examine the functional interactions of TRAK2 with kinesin-1 and dynein–dynactin. We found that TRAK2 robustly activates kinesin-1 for processive transport toward the microtubule plus-end in single-molecule assays using cellular extracts. TRAK2 minimally activates dynein under these same conditions, but expression of exogenous Lissencephaly-1 (LIS1) induces highly processive dynein motility, resulting in significantly more frequent minus-end-directed movement with longer run lengths and higher velocities. Dynein motility is dependent on the conserved CC1-Box dynein adaptor motif within TRAK2, which facilitates an interaction between TRAK2 and dynein. We used co-immunoprecipitation and co-localization experiments to provide evidence for the formation of a complex containing TRAK2, kinesin-1, and dynein–dynactin. Knockdown studies indicate that TRAK2 initiates transport toward either microtubule end more frequently if the opposing motor is present. Together, these results indicate that kinesin-1 and dynein–dynactin are functionally interdependent when in complex with TRAK2, providing mechanistic insight into the coordinated motility of mitochondria within the cell.

## Results

**TRAK2 activates kinesin-1.** To study the functional effects of TRAK2 on kinesin-1 and dynein, we utilized an in vitro single-molecule approach using total internal reflection fluorescence (TIRF) microscopy of cell extracts to characterize microtubule-based motility of individual TRAK2-motor complexes (Fig. 1b)[35–37]. We expressed Halo-tagged TRAK2 in COS-7 cells and labeled cells with tetramethylrhodamine (TMR)-HaloTag ligand prior to generation of cell lysates; the endogenous kinesin, dynein, and dynactin present in COS-7 lysate allowed us to examine the interaction of Halo-TRAK2 with these proteins. Cell

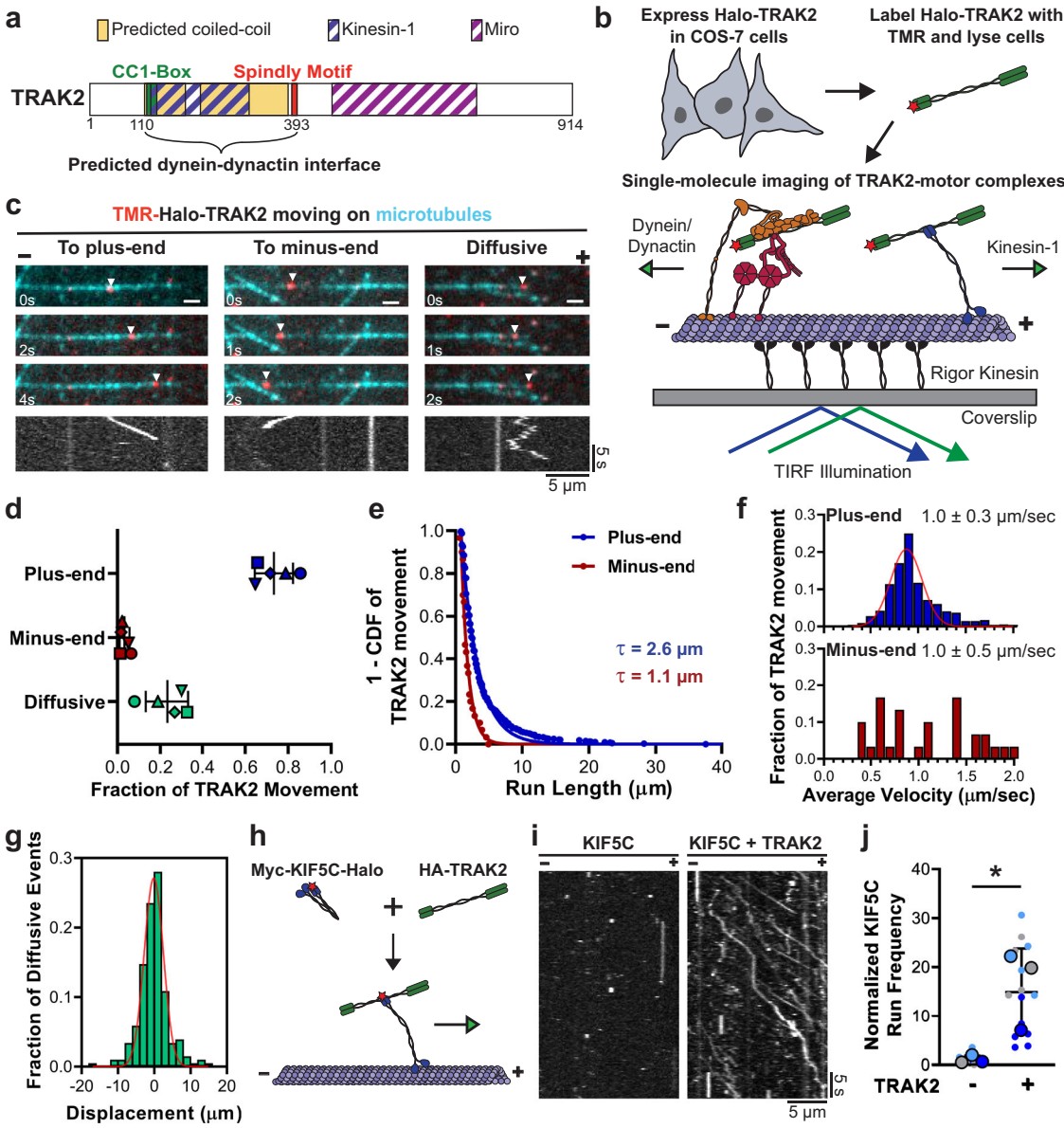

**Fig. 1 TRAK2 activates kinesin-1. a** Diagram of TRAK2 with binding regions for kinesin-1 and Miro, conserved dynein adaptor motifs, and predicted coiled-coil domains. **b** Schematic depiction of the experimental design with an illustration of TMR-labeled Halo-TRAK2 in complex with dynein–dynactin or kinesin-1. **c** Time series and corresponding kymographs showing single Halo-TRAK2 particles (arrow) moving to the plus-end, minus-end, and bidirectionally along dynamic microtubules. White scale bars = 2 μm. **d** Fraction of occurrence for each type of TRAK2 movement. Data points are shaped according to experimental replicate. The center line and bars are the mean ± s.d. from five independent experiments. **e, f** Inverse cumulative distribution functions (CDF) of run length and histogram distributions of velocity for TRAK2 transport to either microtubule end (n = 758 plus-end events and 30 minus-end events). The curves in (**e**) represent a single exponential decay fit, with decay constant indicated above. In **f**, the red curve represents a Gaussian distribution fit. In **f**, the mean ± s.d. is shown. **g** Distribution of run displacement of TRAK2 diffusive movement (n = 203 events). Negative displacement indicates movement to the microtubule minus-end. The red line represents a Gaussian distribution fit. **h** Schematic of KIF5C activation experiment. **i** Representative kymographs showing activation of KIF5C upon co-expression of TRAK2. **j** Normalized frequency of KIF5C motility with and without exogenous TRAK2. Data points are color-coded according to experimental replicate, with smaller points representing KIF5C frequency per video. The center line and bars are the mean ± s.d. from three independent experiments (n = 14 videos without TRAK2 and 11 videos with TRAK2) *p = 0.0320 (two-tailed t-test).

lysates were flowed together with fluorescently labeled tubulin heterodimers into chambers containing immobilized GMPCPP-stabilized microtubule seeds. Chambers were incubated at 37 °C to visualize the movement of TRAK2 on dynamic microtubules. Use of dynamic microtubules more closely models the in vivo environment, and allows for the direct assessment of transport direction due to the greater growth and catastrophe rates of the microtubule plus-end[38–40]. We verified our ability to

unambiguously identify plus- and minus-end-directed motility in control experiments using either purified KIF5B motor head domain (aa 1–560) or the KIF5C motor head expressed in COS-7 cells, both of which exhibited unidirectional transport toward the microtubule plus-end, while a truncated N-terminal construct of the dynein activator HOOK1 (aa 1–554) expressed in COS-7 cells exhibited unidirectional transport toward the microtubule minus-end (Supplementary Fig. 1).

Using this system, we observed three distinct kinds of motility for individual TRAK2-motor complexes: unidirectional transport to the microtubule plus-end, unidirectional transport to the microtubule minus-end, and short back-and-forth movements (Fig. 1c and Supplementary Fig. 2). Unidirectional TRAK2 transport toward either microtubule end was processive, as determined by the parabolic fit from mean-squared displacement (MSD) analysis of these runs (Supplementary Fig. 3a, b). These processive movements are consistent with motor-based transport by kinesin-1 and dynein. The majority of TRAK2 movement was toward the microtubule plus-end (Fig. 1d and Supplementary Movie 1). Plus-end-directed TRAK2 transport was highly processive, with runs up to 38 μm and a mean velocity of 1 μm/s (Fig. 1e, f). TRAK2 transport toward the microtubule minus-end was far less robust than transport toward the plus-end, with only ~3% of TRAK2 runs directed toward the minus-end (Fig. 1d). These minus-end-directed runs were short, with no runs >5 μm, and displayed variable velocities up to 2 μm/s (Fig. 1e, f). These observations suggest that TRAK2 strongly activates kinesin and minimally activates dynein under the conditions of this assay.

The remaining ~23% of TRAK2 motility was bidirectional and characterized by frequent directional switches. These movements could displace TRAK2 up to 20 μm in either direction, but typically resulted in minor positional changes with no bias toward either microtubule end (Fig. 1g). This bidirectional motility is consistent with diffusion of TRAK2 along the microtubule, as shown by the linear fit from MSD analysis of these movements (Supplementary Fig. 3c). Diffusive movements were motor-independent as neither the frequency nor the displacement of these events were affected by knockdown of endogenous KIF5B, dynein heavy chain (DHC), or p150[Glued] (Supplementary Fig. 3d–g). TRAK1 was recently shown to contain a microtubule-binding domain within its C-terminus that allows for diffusion along microtubules in vitro[41]. Given the high similarity between TRAK1 and TRAK2, the motor-independent diffusion of TRAK2 along microtubules is likely facilitated by a direct interaction between TRAK2 and the microtubule.

Our finding that TRAK2 induces processive transport toward the microtubule plus-end suggests that TRAK2 activates kinesin-1, similar to how TRAK1 activates the kinesin-1 isoform KIF5B for processive transport along microtubules in vitro[41]. To directly test if TRAK2 activates kinesin-1 for processive transport toward the microtubule plus-end, we transfected COS-7 cells with Halo- and Myc-tagged KIF5C and HA-tagged TRAK2. We then labeled cells with TMR-HaloTag ligand and performed our single-molecule motility assay on dynamic microtubules in the presence or absence of exogenous HA-TRAK2 (Fig. 1h). When expressed alone, KIF5C rarely displayed movement along microtubules, consistent with the autoinhibitory head-to-tail folding of inactive kinesin-1 (Fig. 1i)[42]. Occasional movements toward the microtubule plus-end were observed, likely due to stochastic activation by endogenous adaptors present at low levels in the lysate (Supplementary Movie 2)[43]. When TRAK2 was co-expressed, the frequency of KIF5C runs increased by more than tenfold (Fig. 1i, j and Supplementary Movie 3). Further, the presence of exogenous TRAK2 promoted longer KIF5C runs toward the microtubule plus-end with slightly increased velocities (Supplementary Fig. 4). Thus, TRAK2 activates kinesin-1 to increase this molecular motor's processivity toward the microtubule plus-end.

**LIS1 promotes processive dynein-mediated TRAK2 movement.** Because we rarely observed minus-end-directed TRAK2 transport under the conditions of this assay, we hypothesized that we might be lacking a component necessary for activation of dynein. One candidate regulator is LIS1, which binds directly to the dynein motor domain[44–46]. LIS1 has been reported to either antagonize or to activate processive dynein motility[47]. However, recent studies found that LIS1 enhances dynein activity by stabilizing an open, uninhibited dynein conformation, promoting the assembly of the motile dynein–dynactin-adaptor complex, and favoring the recruitment of a second dynein motor to the same complex[44,48–50]. Recruitment of a second dynein motor increases the velocity and force production of individual dynein–dynactin-adaptor complexes[29,44,48]. This activating property of LIS1 is found across dynein–dynactin-adaptor complexes, including both CC1-Box-containing adaptors (BICD2 and BICDR1) and structurally unrelated adaptors (Hook3 and Ninl)[44,48], suggesting that LIS1 might similarly activate dynein–dynactin in complex with TRAK2.

To determine the effect of LIS1 on the motility of TRAK2-motor complexes, we expressed Halo-TRAK2 with or without HA-tagged LIS1 in COS-7 cells and examined the motility of individual TRAK2-motor complexes moving along dynamic microtubules with TIRF microscopy (Fig. 2a). Expression of HA-LIS1 raised LIS1 protein levels four- to eightfold above endogenous levels present in cell extracts (Supplementary Fig. 5). We found that increased expression of LIS1 promoted robust TRAK2 transport toward the microtubule minus-end, causing a ~7-fold increase in the frequency of minus-end motility (Fig. 2b, c, Supplementary Fig. 2, and Supplementary Movie 4). This increase in minus-end run frequency coincided with increased processivity of minus-end-directed TRAK2 motility. Minus-end runs were longer with exogenous LIS1 present, with 24% of runs over 5 μm (Fig. 2d). Exogenous LIS1 also caused a significant increase in the velocity of minus-end-directed TRAK2 transport ($p = 0.0008$, two-tailed Mann–Whitney U test), with 23% of runs displaying velocities over 2 μm/s (Fig. 2e). The observation of fast, sustained movements toward the microtubule minus-end upon the addition of LIS1 is consistent with recent reports that LIS1 promotes the formation of dynein–dynactin-adaptor complexes[44,48,49]. We conclude that LIS1 promotes the activation of dynein–dynactin–TRAK2 complexes for processive transport.

Although LIS1 robustly increased the frequency of dynein-mediated TRAK2 transport, we did not observe any effect of LIS1 on the frequency of plus-end-directed or diffusive TRAK2 events (Fig. 2b). Similarly, the run length and velocity of plus-end runs were unaffected by the addition of LIS1, indicating that LIS1 does not affect the processivity of kinesin-1 when in complex with TRAK2 (Fig. 2f, g). These results indicate that LIS1 functions as a dynein-specific activator of TRAK2 transport.

While LIS1 had no effect on plus-end TRAK2 movement, these events still comprised the majority of TRAK2 movement, regardless of whether exogenous LIS1 was present. This bias for kinesin-based motility was surprising given the reported preference of TRAK2 for promoting dynein-based mitochondrial transport within cells[14,22]. In contrast to TRAK2, TRAK1 has been shown to preferentially promote kinesin-based transport of mitochondria[14,22]. As a result, we hypothesized that the observed plus-end bias of TRAK2 in our system might be caused by the presence of endogenous TRAK1 in complex with Halo-TRAK2. To test this possibility, we first examined if TRAK2 interacts with TRAK1 by expressing Halo-TRAK2 with HA-tagged TRAK1 in COS-7 cells. Using an antibody for the Halo tag, we immuno-precipitated Halo-TRAK2 and pulled down HA-TRAK1, confirming that TRAK2 binds TRAK1 (Supplementary Fig. 6a, b).

We next examined whether knockdown of TRAK1 affects TRAK2 transport in our lysate-based single-molecule assay. Knockdown of TRAK1 by siRNA resulted in an ~70% reduction in the level of TRAK1 protein in COS-7 cells (Supplementary Fig. 6c, d). Characterization of Halo-TRAK2 transport along

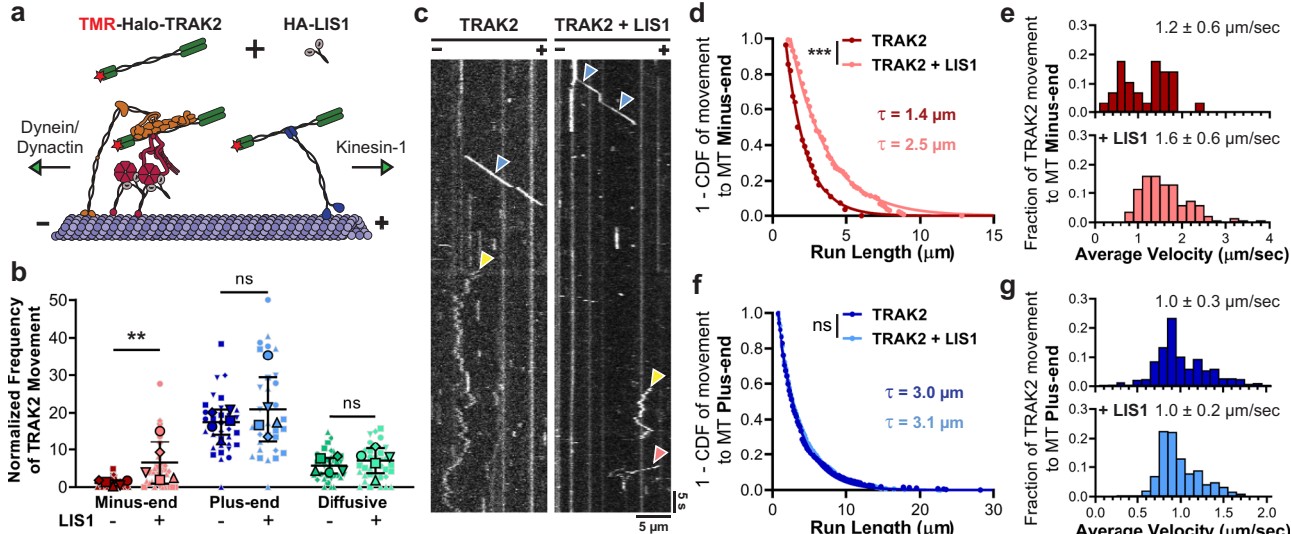

**Fig. 2 LIS1 enhances processive TRAK2 transport to the microtubule minus-end. a** Schematic illustration of experimental design, with LIS1 shown binding the dynein motor domain. **b** Normalized frequency of TRAK2 motile events with and without exogenous HA-LIS1. Data points are shaped according to experimental replicate, with smaller points representing TRAK2 frequency per video. The center line and bars represent the mean ± s.d., ($n = 34$ videos per condition, five independent experiments). **$p < 0.01$; ns not significant (two-tailed Mann–Whitney U test). $p$ values: minus-end, $p = 0.0079$; plus-end, $p = 0.6905$; diffusive, $p = 0.4206$. **c** Representative kymographs showing that LIS1 induces minus-end-directed TRAK2 motility. Red, blue, and yellow arrows indicate minus-end, plus-end, and diffusive TRAK2 movement, respectively. **d–g** Inverse cumulative distribution functions (CDF) of run length and histogram distributions of velocity for TRAK2 transport to either microtubule end with or without HA-LIS1 ($n = 28$ minus-end events without LIS1, 135 minus-end events with LIS1, 369 plus-end events without LIS1, and 570 plus-end events with LIS1) ***$p = 0.0002$; ns not significant, $p = 0.5265$ (two-tailed Mann–Whitney U test). In **d**, **f**, single exponential decay curve fits are shown with decay constants indicated above. The values in (**e**, **g**) are mean ± s.d.

microtubules in lysates prepared from these cells showed no marked changes in the frequency, run length, or velocity of runs toward either microtubule end (Supplementary Fig. 6f–j). Since TRAK2 displays minimal transport toward the microtubule minus-end under these conditions, we performed the same experiment with HA-LIS1 expressed to promote processive TRAK2 transport toward the microtubule minus-end. Even with exogenous LIS1 present, we did not observe any effect of TRAK1 knockdown on the frequency, run length, or velocity of TRAK2 runs toward either microtubule end (Supplementary Fig. 6k–p). In all conditions examined, TRAK2 preferentially promoted transport toward the microtubule plus-end, indicating that the plus-end bias of TRAK2 in this system is not due to the presence of TRAK1 in these motor-adaptor complexes.

**The TRAK2 CC1-Box is required for processive motility to the microtubule minus-end.** The CC1-Box of the TRAK proteins is conserved among many dynein activating adaptors, such as Spindly and BICD2 (Fig. 3a), where it facilitates a direct inter-action with dynein light intermediate chain 1 (LIC1)[30,31,35,51]. Structural work on the BICD2 dimer indicates that the CC1-Box forms a hydrophobic pocket that binds LIC1[31]. This LIC1–adaptor interaction is necessary for processive dynein motility in CC1-Box-containing adaptors and in the Hook family of adaptors, which bind LIC1 through an analogous coiled-coil segment[35]. Within the CC1-Box, multiple residues are essential for the adaptor interaction with dynein. Mutating two conserved alanine residues to valine residues in the CC1-Box of Spindly or BICD2 reduces the interaction with dynein and dynactin[30,52] while mutating a nearby tyrosine residue to aspartic acid in BICD2 disrupts the adaptor interaction with LIC1[31]. Mutating the analogous isoleucine to aspartic acid in HAP1 is sufficient to disrupt dynein-dependent HAP1 motility in neurons[51].

To test if the TRAK2 CC1-Box is necessary for processive dynein-dependent transport to the microtubule minus-end, we introduced point mutations predicted to disrupt the interaction of TRAK2 with dynein: two alanine to valine (A129V, A130V) mutations and an isoleucine to aspartic acid (I132D) mutation. Isoleucine residue 132 is analogous to the tyrosine residue that is required for BICD2 to interact with LIC1, suggesting that it facilitates an interaction with LIC1[31]. We then compared the motility of Halo-TRAK2 with A/V or I/D mutations to wild type (WT) in TIRF; we co-expressed HA-LIS1 with Halo-TRAK2 to promote transport to the microtubule minus-end. Both CC1-Box mutations caused a marked reduction in the frequency of processive runs toward the microtubule minus-end (Fig. 3b, c). TRAK2 I/D displayed reduced run lengths and velocities when moving to the minus-end while TRAK2 A/V only displayed shortened run lengths (Fig. 3d, e). These impaired minus-end runs resembled those of TRAK2 without activation by exogenous LIS1, indicating that the TRAK2 CC1-Box is required to activate dynein for robust transport to the microtubule minus-end. Surprisingly, these CC1-Box mutations also caused a slight reduction (33% for A/V and 42% for I/D) in the frequency of TRAK2 runs toward the microtubule plus-end (Fig. 3f). However, neither CC1-Box mutation altered the run length or velocity of plus-end-directed TRAK2 motility, indicating that the CC1-Box is not required to activate kinesin-1 for processive transport to the microtubule plus-end. (Fig. 3g, h).

To determine whether the CC1-Box affects binding of TRAK2 to dynein and kinesin-1, we expressed WT, A/V, or I/D versions of Halo-TRAK2 with Myc-tagged KIF5B in COS-7 cells and immunoprecipitated with an anti-Halo antibody. We observed that both CC1-Box mutations reduced TRAK2 binding to endogenous DHC (Fig. 3i, j), consistent with the low number of dynein runs initiated by TRAK2 A/V and I/D. In contrast, neither mutation had any effect on the interaction between

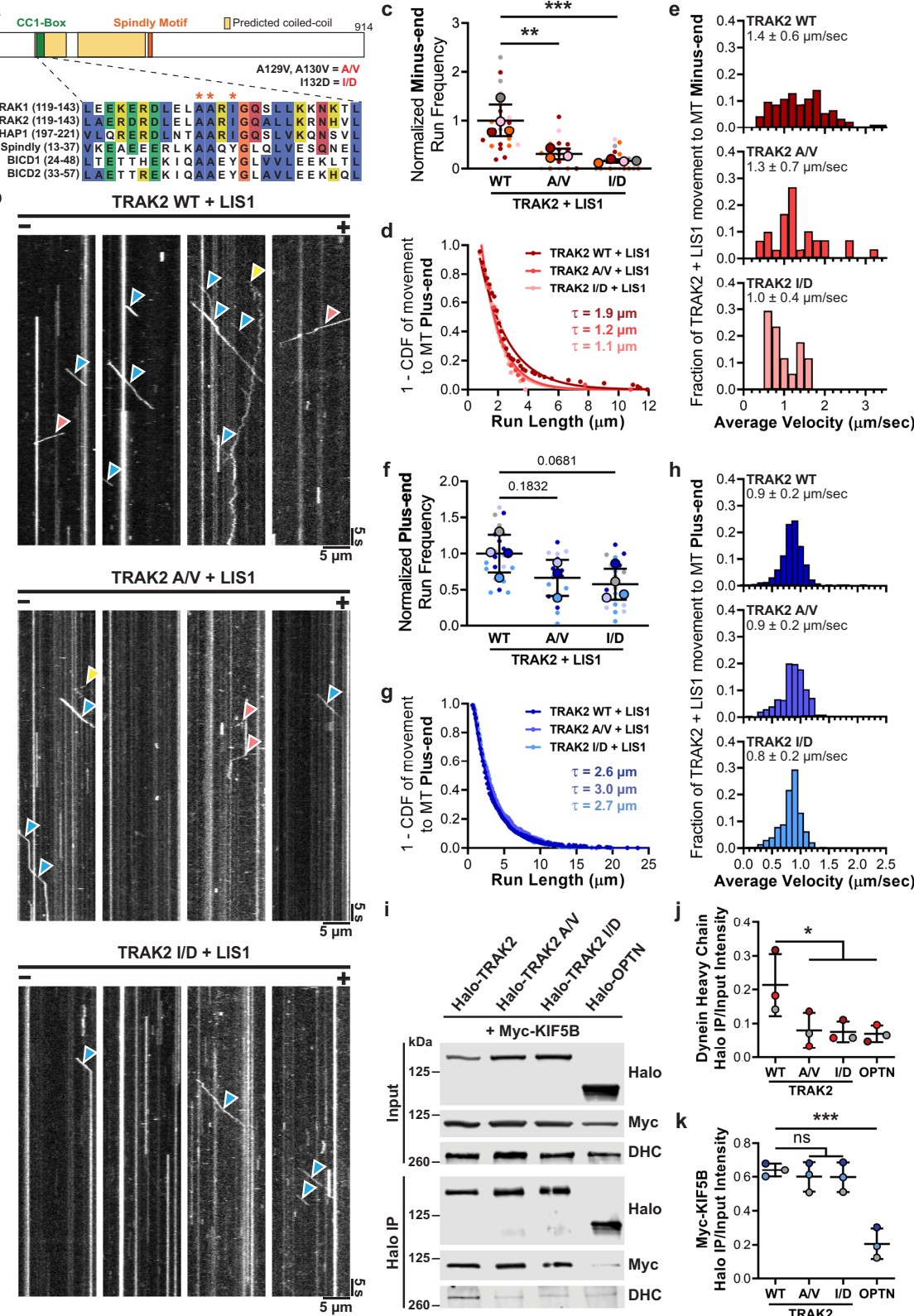

TRAK2 and kinesin-1 (Fig. 3i, k). Combined, these results indicate that the TRAK2 CC1-Box specifically facilitates an interaction with dynein to promote dynein-mediated transport.

**Kinesin-1 and dynein–dynactin promote TRAK2 transport by the opposing motor**. We next sought to determine the relationship between kinesin-1 and dynein–dynactin within single

TRAK2-motor complexes. We used siRNAs against KIF5B, DHC, or p150[Glued] to knock down endogenous kinesin-1, dynein, or dynactin, respectively, within COS-7 cells expressing Halo-TRAK2. We then used these cell lysates to study the microtubule-based transport of TRAK2 in TIRF; this experiment was performed with either cells expressing just Halo-TRAK2 or cells co-expressing HA-LIS1 as a way to promote transport to the

**Fig. 3 The TRAK2 CC1-Box is important for processive motility and binding to dynein, but not kinesin-1. a** Schematic overview of TRAK2 with sequence alignment showing conservation of the CC1-Box. The red stars and text indicate the mutations introduced to TRAK2. **b** Representative kymographs showing the effect of A/V and I/D mutations on TRAK2 motility along MTs when LIS1 is also expressed. Red, blue, and yellow arrows indicate minus-end, plus-end, and diffusive TRAK2 movement, respectively. **c** Normalized frequency of TRAK2 transport to the microtubule minus-end with exogenous LIS1. Data points are color-coded to experimental replicate, with smaller points representing TRAK2 frequency per video. The center line and bars represent the mean ± s.d. from independent experiments ($n = 22$ videos from four experiments for WT, 18 videos from three experiments for A/V, and 21 videos from four experiments for I/D). $**p = 0.0047$; $***p = 0.0009$ (one-way ANOVA with the Dunnett's multiple comparisons test). **d**, **e** Inverse cumulative distribution functions (CDF) of run length and histogram distributions of velocity for TRAK2 transport to the microtubule minus-end with exogenous LIS1 present ($n = 92$ events for TRAK2 WT, 30 events for TRAK2 A/V, and 17 events for TRAK2 I/D). The curves in (**d**) represent single exponential decay fits with decay constants indicated above. The values in (**e**) are mean ± s.d. **f** Same as **c**, but for TRAK2 transport to the microtubule plus-end. Exact $p$ values from one-way ANOVA with the Dunnett's multiple comparisons test are shown. **g**, **h** Same as **d**, **e**, but for TRAK2 transport to the microtubule plus-end ($n = 693$ events for TRAK2 WT, 369 events for TRAK2 A/V, and 391 events for TRAK2 I/D). **i** Immunoprecipitation using a Halo antibody of extracts from COS-7 cells transfected with Myc-KIF5B and Halo-tagged TRAK2, TRAK2 A/V, TRAK2 I/D, or negative control Optineurin (OPTN). **j** Quantification of co-immunoprecipitation of endogenous dynein heavy chain (DHC) with Halo-tagged constructs. Data points are color-coded according to experimental replicate. The center line and bars represent the mean ± s.d. from three independent experiments. $*p < 0.05$ (one-way ANOVA with the Dunnett's multiple comparisons test). $p$ values: WT vs. A/V, $p = 0.0475$; WT vs. I/D, $p = 0.0416$; WT vs. OPTN, $p = 0.0346$. **k** Same as **j** but for co-immunoprecipitation of Myc-KIF5B. $***p < 0.001$; ns not significant (one-way ANOVA with the Dunnett's multiple comparisons test). $p$ values: WT vs. A/V, $p = 0.8697$; WT vs. I/D, $p = 0.8513$; WT vs. OPTN, $p = 0.0004$.

microtubule minus-end. Using this system, we found that knockdown of KIF5B significantly reduced the frequency of plus-end-directed TRAK2 transport by 94% and 85%, with and without LIS1, respectively (Fig. 4a, b and Supplementary Fig. 7a, b). This result confirms that the observed plus-end motility of TRAK2 is driven by kinesin-1 and supports previous reports that kinesin-1 is the primary motor driving mitochondrial transport to the microtubule plus-end[13,14]. Surprisingly, knockdown of dynein or dynactin, with LIS1 present, reduced the frequency of TRAK2 transport toward the plus-end by 44 and 51%, respectively (Fig. 4a, b). Knockdown of dynein or dynactin also caused a slight increase in the run length of TRAK2 transport toward the microtubule plus-end, but had no effect on the velocity of these plus-end runs (Fig. 4c, d). Knockdown of dynein and dynactin had similar effects on the frequency, run length, and velocity of plus-end plus-end-directed TRAK2 transport without exogenous LIS1 present (Supplementary Fig. 7a–d). Thus, the initiation of kinesin-dependent TRAK2 transport to the microtubule plus-end is enhanced by dynein and dynactin, independent of dynein activation via LIS1.

We also examined the effect of motor knockdown on TRAK2 transport toward the microtubule minus-end. This assessment of minus-end transport was done with and without expression of exogenous LIS1, but the rarity of minus-end-directed TRAK2 transport without LIS1 precluded meaningful comparison between conditions (Supplementary Fig. 7e–g). With exogenous LIS1 present, knockdown of dynein or dynactin reduced the frequency of minus-end motility by 94% and 82%, respectively (Fig. 4e), as expected for dynein-mediated motility. However, KIF5B knockdown also reduced the frequency of minus-end-directed TRAK2 events by 58% (Fig. 4e). Despite this large decrease in dynein motility upon knockdown of KIF5B, we observed only a small decrease in run length and no change in velocity of minus-end-directed TRAK2 transport upon KIF5B knockdown (Fig. 4f, g). Thus, kinesin-1 specifically enhances the initiation of TRAK2 dynein motility.

We wondered if the reduction in TRAK2 transport upon motor knockdown was due to reduced association of TRAK2 with microtubules. To test this, we expressed HA-TRAK2 in COS-7 cells, knocked down endogenous KIF5B, DHC, or p150[Glued], and examined the ability of HA-TRAK2 to pellet with GMPCPP-stabilized microtubules. Under control conditions, TRAK2 pellets with microtubules (Fig. 4h, i). Knockdown of KIF5B, DHC, or p150[Glued] was sufficient to reduce the ability of TRAK2 to pellet with microtubules, indicating that each component contributes to

the association of TRAK2 with microtubules. Combined, these data show that kinesin-1, dynein, and dynactin promote the association of TRAK2 with microtubules and promote initiation of processive TRAK2 transport toward either microtubule end.

**TRAK2 facilitates an interaction between kinesin-1 and dynein–dynactin.** Our finding that kinesin-1 and dynein–dynactin promote the initiation of TRAK2 transport by the opposing motor raised the question of whether TRAK2 concurrently interacts with these motors. It is unclear whether TRAK2 interacts with these motors alternately or simultaneously, but our knockdown studies suggest that these motors cooperate within single TRAK2-motor complexes to initiate processive movement. In line with these findings, we observed instances of TRAK2 switching directions during a run while examining TRAK2 transport in the presence of LIS1. These events consisted of processive movement with a single immediate change in direction and were typically accompanied by a change in velocity upon directional switch (Fig. 5a and Supplementary Movie 5). Switches could occur in either direction, but the majority (85%) of switches were in the plus-to-minus direction. We interpret these plus-to-minus directional switches in the presence of LIS1 as transient activation of dynein in TRAK2-motor complexes containing kinesin-1. To further test the possibility that LIS1 activates dynein to promote minus-end-directed transport of motor complexes containing TRAK2 and kinesin-1, we expressed Myc-KIF5C-Halo and HA-TRAK2 with or without HA-LIS1 in COS-7 cells and used TIRF microscopy to track the movement of activated KIF5C along dynamic microtubules. Without exogenous LIS1 present, KIF5C moved unidirectionally toward the microtubule plus-end, as expected. In the presence of LIS1, we found several instances of KIF5C moving processively toward the microtubule minus-end (Supplementary Fig. 8). Together, these data suggest that both kinesin-1 and dynein–dynactin can form a complex with TRAK2 and that the activities of these motors are coordinately regulated to achieve changes in direction.

To directly test if TRAK2 can simultaneously bind kinesin-1 and dynein–dynactin, we performed two immunoprecipitation experiments. We first tested if TRAK2 can form a complex with kinesin-1 and dynactin by expressing HA-tagged TRAK2 alongside Myc-tagged KIF5B and a FLAG-tagged p150 subunit of dynactin in COS-7 cells. Using an antibody for the FLAG tag, we immunoprecipitated FLAG-p150 and pulled down both HA-TRAK2 and Myc-KIF5B, confirming the formation of a TRAK2/kinesin-1/dynactin motor complex (Fig. 5b). When the same

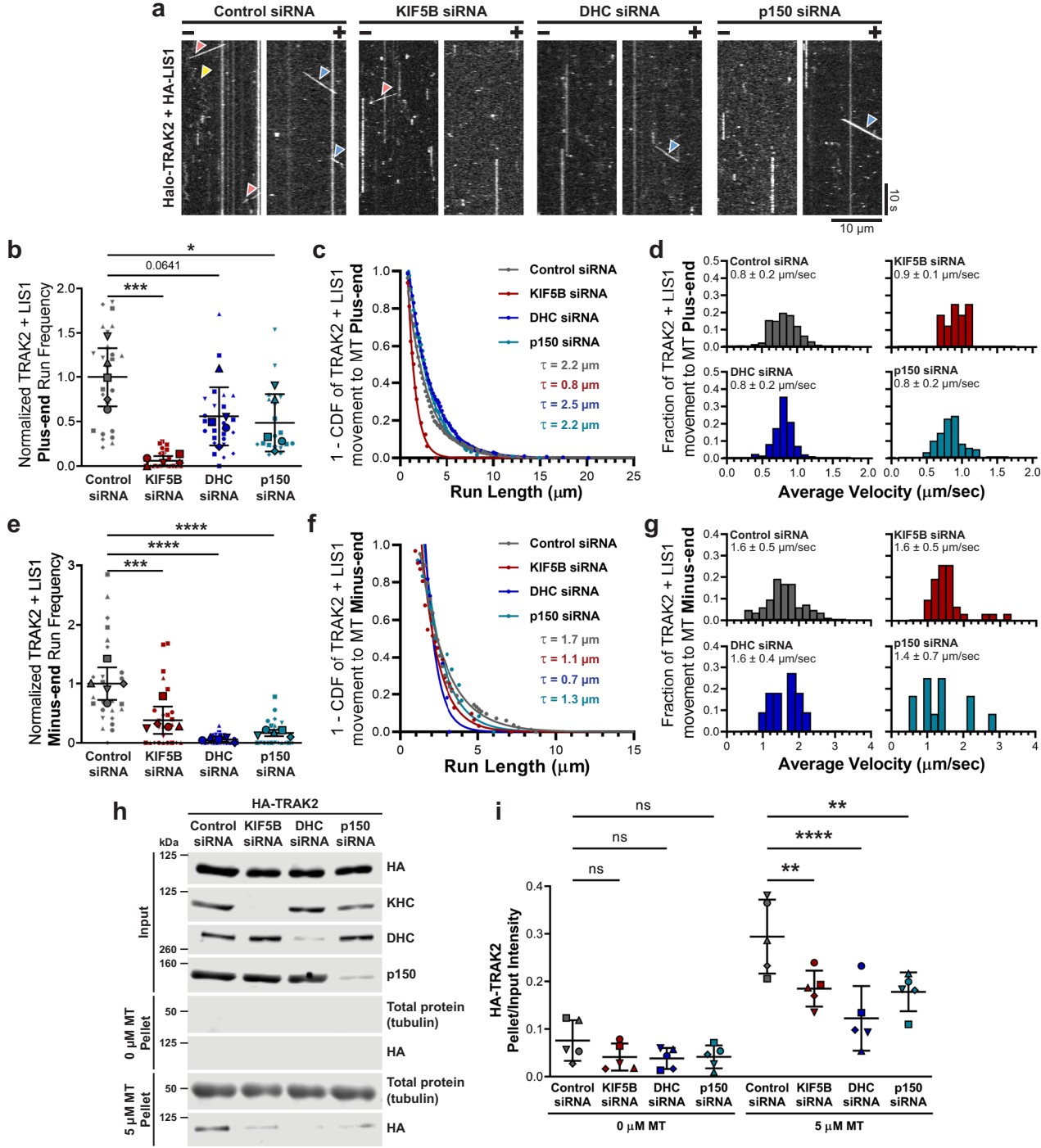

immunoprecipitation was performed in cells expressing just FLAG-p150 and Myc-KIF5B, less Myc-KIF5B was pulled down, indicating that TRAK2 enhances the association between kinesin-1 and dynactin (Fig. 5b, c). Next, we tested if TRAK2 can form a complex with kinesin-1 and dynein by expressing HA-tagged TRAK2 alongside Myc- and Halo-tagged KIF5C in COS-7 cells. Using a Halo antibody, we immunoprecipitated KIF5C and pulled down both HA-TRAK2 and endogenous DHC, confirming the formation of a TRAK2/kinesin-1/dynein motor complex (Fig. 5d). When the same immunoprecipitation was performed in cells expressing Myc-KIF5C-Halo alone, significantly less DHC coimmunoprecipitated with KIF5C, indicating that TRAK2 enhances the association between kinesin-1 and dynein (Fig. 5b, c).

To further verify that TRAK2 can concurrently bind kinesin-1 and dynein–dynactin we used TIRF microscopy to perform co-localization experiments. We first tested whether TRAK2 can form a motile complex with kinesin and dynactin present by expressing Halo-TRAK2 alongside a GFP-tagged p25 subunit of the dynactin pointed-end complex in COS-7 cells. We then performed dual-color TIRF imaging of TMR-labeled Halo-TRAK2 and GFP-p25 to track the motility of co-complexes on dynamic microtubules. As expected, we observed co-localization and co-migration of TRAK2-dynactin complexes (Supplementary Fig. 9a, b). These co-complexes displayed both processive and diffusive movements along the microtubule, as seen previously for TRAK2. Moreover, we observed TRAK2-dynactin complexes moving processively toward the microtubule plus-end, indicating

**Fig. 4 Kinesin-1 and dynein–dynactin promote TRAK2 transport by the opposing motor. a** Representative kymographs showing how siRNA knockdown of KIF5B, dynein heavy chain (DHC), or p150[Glued] (p150) affects TRAK2 motility along MTs when LIS1 is expressed. Red, blue, and yellow arrows indicate minus-end, plus-end, and diffusive TRAK2 movement, respectively. **b** Normalized frequency of TRAK2 transport to the MT plus-end upon motor knockdown when LIS1 is expressed. Data points represent the frequency of TRAK2 motility per video normalized to the average frequency of control siRNA events. The center line and bars represent mean ± s.d., ($n = 26$ videos for control siRNA, 26 for KIF5B siRNA, 25 for DHC siRNA, and 21 for p150 siRNA, five independent experiments). Exact $p$ values are shown when $p > 0.05$; *$p = 0.029$; ***$p = 0.0002$ (one-way ANOVA with the Dunnett's multiple comparisons test). **c,d** Inverse cumulative distribution functions (CDF) of run length and histogram distributions of velocity for TRAK2 transport to the microtubule plus-end with exogenous LIS1 present ($n = 301$ events for control siRNA, 16 events for KIF5B siRNA, 302 events for DHC siRNA, and 121 events for p150 siRNA). The curves in (**c**) represent single exponential decay fits with decay constants indicated above. The values in (**d**) are mean ± s.d. **e** Same as **b**, but for TRAK2 transport to the microtubule minus-end. ***$p = 0.0002$; ****$p < 0.0001$ (one-way ANOVA with the Dunnett's multiple comparisons test). **f,g** Same as **c,d** but for TRAK2 transport to the microtubule minus-end ($n = 83$ events for control siRNA, 31 events for KIF5B siRNA, 11 events for DHC siRNA, and 12 events for p150 siRNA). **h** Microtubule-binding assays were performed using cell lysates from HA-TRAK2 transfected COS-7 cells with siRNA knockdown of KIF5B, DHC, p150, or a control siRNA. Lysates were probed for HA, kinesin heavy chain (KHC), dynein heavy chain (DHC), and p150[Glued] (p150). Lysates were incubated with microtubules, spun down, and the resulting microtubule pellets were probed for total protein (tubulin) and HA. **i** Quantification of relative HA-TRAK2 in 0 and 5 µM microtubule pellets from **h**. The center line and bars represent mean ± s.d. from five independent experiments. ns not significant; **$p < 0.01$; ****$p < 0.0001$ (one-way ANOVA with the Dunnett's multiple comparisons test). $p$ values for 0 µM MT conditions: Control siRNA vs. KIF5B siRNA, $p = 0.7476$; Control siRNA vs. DHC siRNA, $p = 0.6789$; Control siRNA vs. p150 siRNA, $p = 0.7551$. $p$ values for 5 µM MT conditions: Control siRNA vs. KIF5B siRNA, $p = 0.0049$; Control siRNA vs. DHC siRNA, $p < 0.0001$; Control siRNA vs. p150 siRNA, $p = 0.0026$.

that kinesin-1 was present and active in these complexes (Supplementary Fig. 9c). The movement of these co-complexes closely resembled that of TRAK2 alone, as most TRAK2-p25 motility consisted of processive transport toward the microtubule plus-end. This bias toward the microtubule plus-end suggests that TRAK2 preferentially promotes kinesin-based motility under the conditions of this assay, even when dynactin is bound to TRAK2. Next, we performed three-color single-molecule imaging of TRAK2, kinesin-1, and dynein (Fig. 5f). For this experiment, we expressed SNAP-tagged TRAK2, Myc-KIF5C-Halo, and HA-LIS1 in HeLa cells stably expressing GFP-tagged dynein heavy chain (DHC-GFP)[53,54]. We then labeled cells with JF646-SNAP ligand and TMR-HaloTag ligand prior to generation of cell lysates and flowed lysates into chambers containing unlabeled microtubules. We observed TRAK2, KIF5C, and DHC move processively along microtubules as a single complex (Fig. 5g and Supplementary Movie 6). Surprisingly, we found that TRAK2-KIF5C-DHC co-complexes exclusively exhibited transport toward the microtubule plus-end (41 of 41 runs containing all three components; Supplementary Fig. 10), providing additional evidence that kinesin-1 functions as the dominant motor when in complex with TRAK2, under the conditions of our assay, regardless of the association of dynein–dynactin. Combined, these results demonstrate that TRAK2 promotes an interaction between kinesin-1 and dynactin–dynactin that allows these opposing motors to be transported together as a multi-motor complex.

## Discussion

TRAK proteins play an essential role as motor adaptors in the transport of mitochondria along microtubules[55,56]. Although specific roles have been proposed for TRAK1 and TRAK2 in the control of mitochondrial transport, our understanding of the molecular mechanisms by which TRAK proteins function is largely unknown. Previous studies of mitochondrial transport in the axons and dendrites of rat hippocampal and cortical neurons identified TRAK1 as the motor adaptor primarily responsible for axonal mitochondrial transport, while dendritic transport depends on the motor adaptor TRAK2[14,57]. It was proposed that these compartmental differences were the result of differential interactions of TRAK1 and TRAK2 with microtubule motors; TRAK1 was proposed to interact with kinesin-1 and dynein–dynactin whereas TRAK2 was thought to primarily interact with dynein–dynactin[14]. However, in our functional studies of single TRAK2-motor complexes, we found that

TRAK2 strongly activates kinesin-1 over dynein. Such a bias against dynein motility could either be due to the absence of dynein from these complexes or insufficient activation of the dynein motor under these assay conditions. We show that TRAK2 robustly activates kinesin-1 irrespective of whether dynein or dynactin is present in the complex, ruling out the possibility that the absence of dynein–dynactin drives this bias. Thus, the reported preference of TRAK2 for binding dynein–dynactin and promoting minus-end-directed transport is likely contextual and dependent on additional factors that regulate motor binding and activity. TRAK2 requires Miro1, but not Miro2, to promote mitochondrial transport toward the microtubule minus-end, suggesting that the Miro isoform bound to TRAK2 may regulate the directional preference of TRAK2-motor complexes[22].

TRAK2 has been predicted to function as an activating adaptor that increases dynein processivity because of its known interaction with dynein–dynactin and the identification of conserved motifs present in well-characterized activating adaptors[27,58]. Interestingly, we found that LIS1 enhances the frequency, run length, and velocity of TRAK2 transport toward the microtubule minus-end, which closely resembles the effect of LIS1 on other dynein activating adaptors: BICD2, Hook3, and Ninl[44,48]. For these dynein activating adaptors, LIS1 transiently binds dynein to promote the formation of activated dynein–dynactin-adaptor complexes. Binding of LIS1 increases the force production and velocity of individual dynein–dynactin-adaptor complexes by recruiting a second dynein dimer. Thus, the fast and sustained minus-end-directed TRAK2 motility induced by LIS1 over-expression is likely due to both the enhanced formation of a dynein–dynactin-adaptor complex and the recruitment of a second dynein dimer. Perhaps the addition of a second dynein dimer allows for dynein to more efficiently initiate processive TRAK2 transport toward the microtubule minus-end.

The processive minus-end-directed motility of TRAK2 induced by LIS1 also depends on the TRAK2 CC1-Box. We find that both the A/V and I/D mutations drastically reduced the frequency of minus-end runs and reduced the interaction between TRAK2 and dynein (Fig. 3). Since these CC1-Box mutations disrupt the interaction between BICD2 and LIC1, we propose that the TRAK2 CC1-Box facilitates a similar interaction with dynein light intermediate chain. Our findings substantiate the role of TRAK2 as a dynein activating adaptor, since minus-end-directed TRAK2 motility is dependent on dynein and dynactin, enhanced by the addition of LIS1, and requires the TRAK2 CC1-Box.

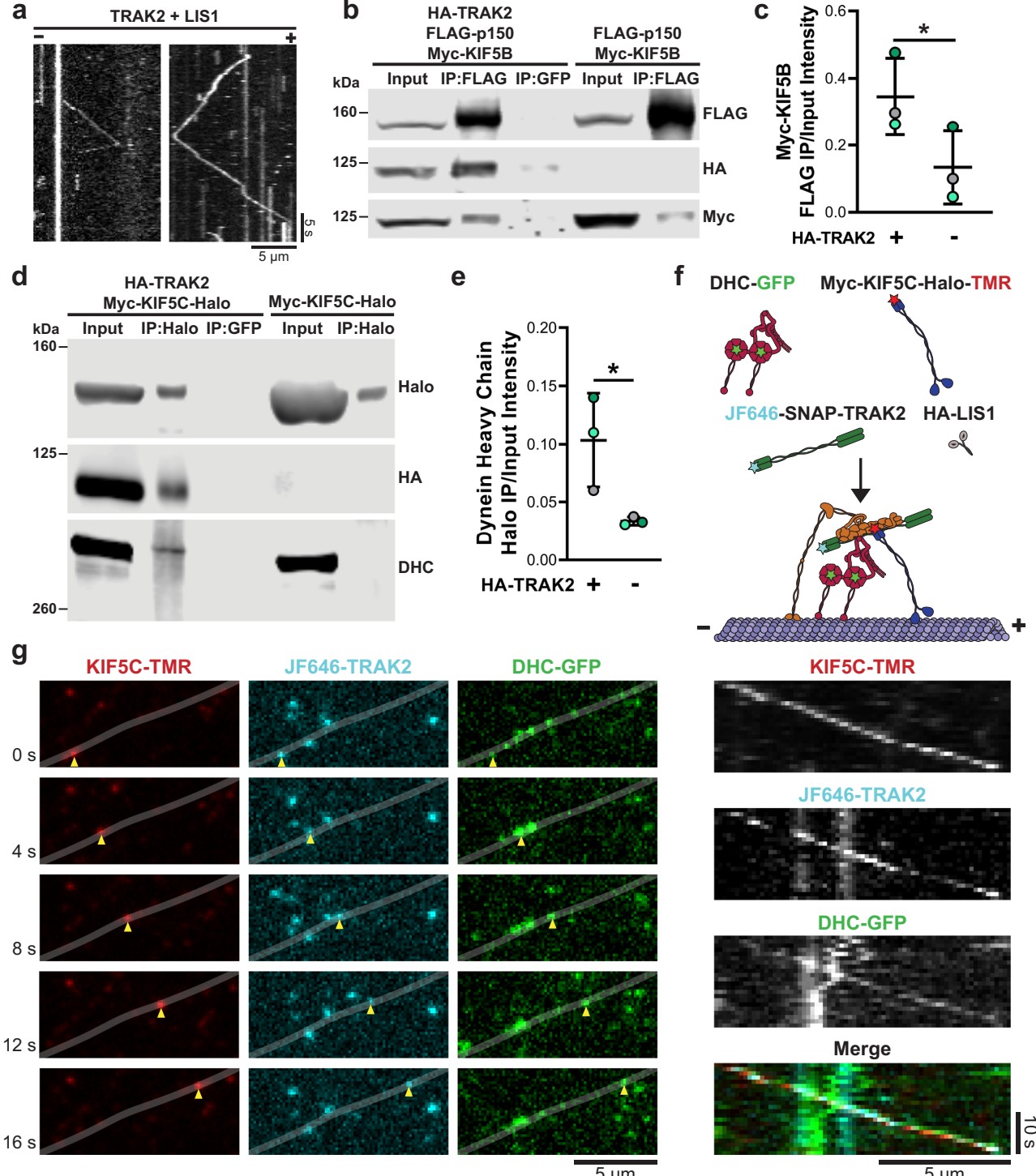

**Fig. 5 TRAK2 forms a complex with kinesin-1, dynein, and dynactin. a** Representative kymographs show TRAK2-motor complexes switching direction during a run. **b** Lysates from COS-7 cells transfected with HA-TRAK2, FLAG-p150[Glued], and Myc-KIF5B were immunoprecipitated with a FLAG antibody or negative control GFP antibody. Lysates from COS-7 cells expressing just FLAG-p150[Glued] and Myc-KIF5B were immunoprecipitated with a FLAG antibody in parallel. **c** Quantification of the difference in Myc-KIF5B co-immunoprecipitation with FLAG-p150[Glued] upon addition of HA-TRAK2. Data points are colored according to experimental replicate. Bars represent mean ± s.d. from three independent experiments. $*p = 0.0405$ (two-tailed $t$-test). **d** Lysates from COS-7 cells transfected with HA-TRAK2 and Myc-KIF5C-Halo were immunoprecipitated with a Halo antibody or negative control GFP antibody. Lysates from COS-7 cells expressing just Myc-KIF5C-Halo were immunoprecipitated with a Halo antibody in parallel. **e** Quantification of the difference in dynein heavy chain co-immunoprecipitation with Myc-KIF5C-Halo upon addition of HA-TRAK2. Data points are colored according to experimental replicate. Bars represent mean ± s.d. from three independent experiments. $*p = 0.0410$ (two-tailed $t$-test). **f** Schematic illustration for three-color single-molecule imaging of GFP-dynein heavy chain, Myc-KIF5C-Halo labeled with TMR, and SNAP-TRAK2 labeled with Janelia Fluor 646. HA-LIS1 is also present in this experiment. **g** Time series showing a processive complex containing KIF5C, TRAK2, and dynein heavy chain. The gray line indicates microtubule position, as inferred from the max projection of KIF5C. Right: corresponding kymographs.

Intriguingly, the predicted dynein–dynactin interface encompasses the proposed kinesin-1-binding region on TRAK2 (Fig. 1a). A similar overlap of kinesin-1 and dynactin-binding regions is present in JIP1, a motor adaptor that regulates the transport of autophagosomes[51,59]. JIP1 is thought to regulate motor activity by switching between mutually exclusive complexes containing either kinesin-1 or dynein–dynactin[43], raising the possibility that TRAK2-motor complexes are mutually exclusive as well. However, we demonstrate here that kinesin-1 and dynein–dynactin can concurrently interact with TRAK2 and that TRAK2 promotes the joining of these opposing motor components (Fig. 5). The ability of TRAK2 to form a motor-adaptor complex with kinesin and dynein may represent a more general feature of bidirectional motor adaptors as the dynein activating adaptor HOOK3 has been shown to scaffold an interaction between dynein–dynactin and the kinesin-3 motor KIF1C for processive transport toward either microtubule end[60]. While both TRAK2 and HOOK3 scaffold kinesin and dynein motors, the relationship between these opposing motors on TRAK2 is unique. We found that genetic depletion of KIF5B, DHC, or the p150[Glued] subunit of dynactin reduced the association of TRAK2 with microtubules and the frequency of TRAK2 transport to both microtubule ends, demonstrating that the functions of these motors are coordinated to promote processive TRAK2 transport along the microtubule (Fig. 4). The frequency of kinesin-based TRAK2 transport was reduced by targeted CC1-Box mutations that specifically disrupt TRAK2 binding to dynein without affecting its interaction with kinesin-1 (Fig. 3f–k), further supporting the notion of a functional linkage between these motors on TRAK2. Moreover, we found that TRAK2-motor complexes could quickly switch from processive transport by one motor to processive transport by the opposing motor. Such immediate directional switches suggest that the activities of kinesin and dynein are tightly regulated within multi-motor TRAK2 complexes to achieve directed transport. Combined, our data support a model in which TRAK2 coordinates kinesin-1 and dynein–dynactin as an interdependent motor complex, providing integrated control of opposing motors (Fig. 6a).

Our findings provide new insights into the control of microtubule-based bidirectional mitochondrial transport. We show that single TRAK2-motor complexes display distinct modes of bidirectional transport that closely resemble the transport of mitochondria in cells. First, we consistently observed motor-independent diffusion of TRAK2 that is characterized by frequent directional switches and no net displacement along the microtubule. These bidirectional movements resemble the short, back-and-forth movements displayed by neuronal mitochondria, which have often been interpreted as an unregulated tug-of-war between opposing motors[1]. Although the exact nature of the bidirectional movement seen in vivo remains uncertain, our findings for single TRAK2-motor complexes are not consistent with a tug-of-war between kinesin and dynein motors, as this bidirectional diffusive motility is largely unaffected by loss of kinesin-1, dynein, or dynactin. Instead, we propose that these short bidirectional movements are the result of one-dimensional diffusion along the microtubule that is likely mediated by TRAK2 itself.

In contrast to this diffusive motility, neuronal mitochondria display a distinct kind of bidirectional transport in which they rapidly reverse their direction of transport during processive movement. The immediate nature of these reversals suggests that kinesin and dynein are simultaneously bound and coordinately regulated on a single mitochondrion. Our observation of immediate directional switching during the processive movement of a TRAK2-motor complex mirrors this behavior of neuronal mitochondria. We propose that mitochondrial transport is facilitated by multi-motor TRAK2 complexes, which allow for integrated control of opposing kinesin and dynein motors (Fig. 6b). The direction of mitochondrial transport could then be regulated by specifically modulating the activities of kinesin and dynein within this multi-motor TRAK complex. One potential direction-specific transport effector is Disrupted-In-Schizophrenia 1 (DISC1). DISC1 interacts with the TRAK/Miro complex to promote kinesin-1-dependent anterograde axonal mitochondrial transport in mouse hippocampal neurons[61]. Another transport effector, NudE neurodevelopmental protein 1, associates with the TRAK/Miro complex to selectively promote retrograde mitochondrial transport[62,63]. The association of these transport effectors, combined with the specific TRAK and Miro isoforms in complex with microtubule motors on mitochondria could then allow for local regulation of motor activity in response to cellular signals.

Our finding that kinesin-1 and dynein–dynactin are functionally linked when in complex with TRAK2 provides insight into the coordination of these motors for mitochondrial transport. It has been apparent that specific disruptions of kinesin-1, dynein, or dynactin cause bidirectional transport defects since the discovery that these motors are essential for fast axonal transport[33]. Initial observations of bidirectional mitochondrial transport in axons similarly found that transport in both directions was affected by mutations in kinesin-1, dynein, or dynactin[13]. This finding that disrupting one motor leads to diminished mitochondrial motility in both directions has been observed across systems[14,32,34,64], raising the question of how opposing microtubule motors are interdependent for mitochondrial transport. We observed a similar interdependence of opposing motors in our single-molecule studies of TRAK2 (Fig. 4), indicating that motor co-dependence is inherent to the TRAK2-motor complex. This observation suggests that the paradox of motor co-dependence for mitochondrial transport stems from molecular interactions within individual TRAK-motor complexes. Loss or inhibition of kinesin-1 or dynein–dynactin might reduce the frequency of transport by the opposing motor within each TRAK-motor complex, reducing mitochondrial transport in both directions. Kinesin-1, dynein, and dynactin each contain a microtubule-binding domain that may facilitate increased association of the TRAK-motor complex with the microtubule, resulting in more frequent transport in either direction. However, an alternative mechanism may account for the paradox of motor co-dependence. TRAK2 is known to undergo conformational changes that control motor binding[14]. These conformational changes may facilitate specific interactions between kinesin-1 and dynein–dynactin that result in a functionally interdependent motor complex. Moreover, the presence of additional factors may be required to functionally link these opposing motors. Our lysate-based approach may include cellular factors that facilitate such an interaction. Future studies of TRAK-motor complexes using purified proteins and in vitro reconstitution of mitochondrial transport will help to elucidate the precise interactions between microtubule motors and TRAK proteins that allow for proper transport and positioning of mitochondria within a cell.

## Methods

**COS-7 cell culture and transfections**. COS-7 (ATCC, CRL-1651) and HeLa cells stably expressing DHC-GFP (gift from A. Hyman, Max Planck Institute for Molecular Cell Biology and Genetics) were cultured in DMEM with 2 mM glutamax (GIBCO) and 10% fetal bovine serum. Cells were transiently transfected with Fugene 6 (Roche) according to manufacturer's instructions and harvested 18–24 h post transfection. The following DNA constructs were used: HA-TRAK2 (gift from C. Hoogenraad, Utrecht University), Halo-TRAK2 (subcloned from HA-TRAK2 into pFN21A-HaloTag-CMV vector from Promega), Halo-TRAK2 AA129-130VV and Halo-TRAK2 I132D (generated from Halo-TRAK2), HA-LIS1 (gift from D.

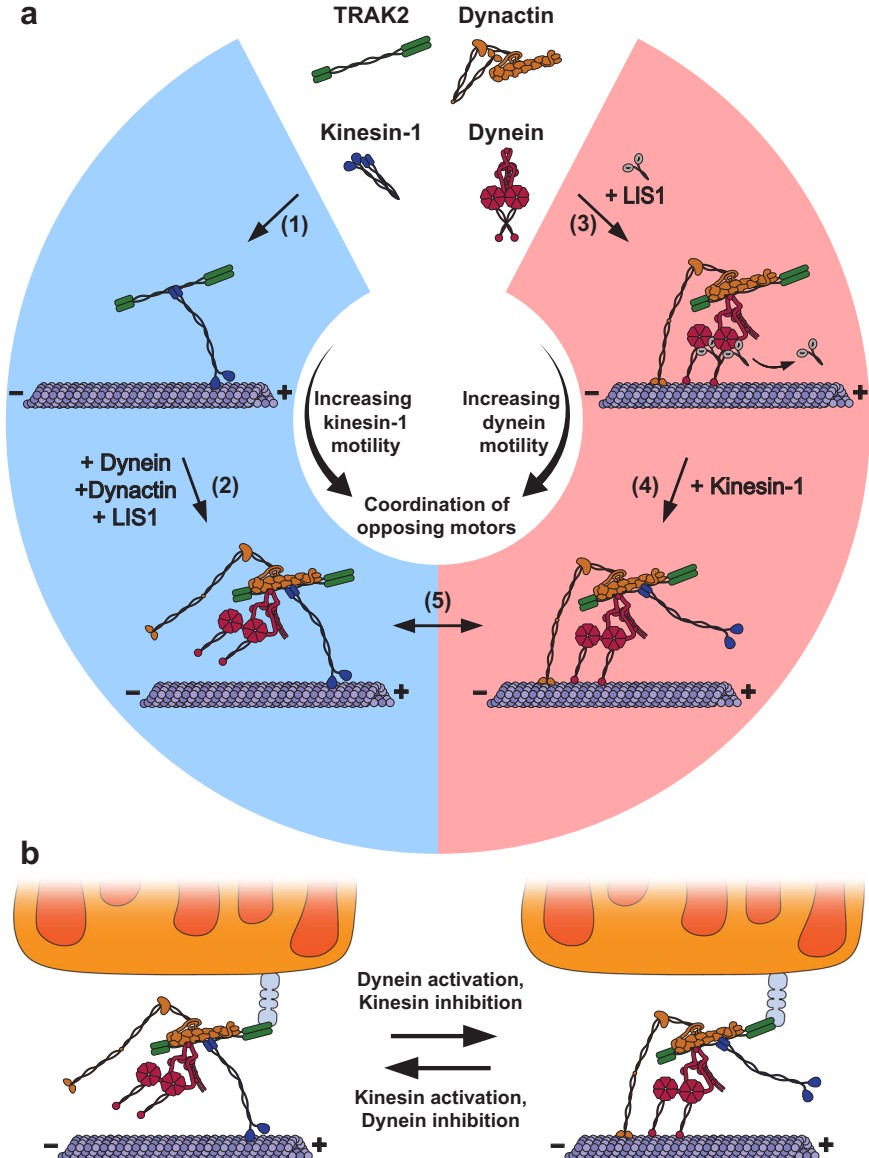

**Fig. 6 TRAK2 functionally links opposing microtubule motors. a** A model for TRAK2 transport by microtubule motors. Blue indicates active kinesin-1 and red indicates active dynein. (1) TRAK2 binds to kinesin-1 to form a motor complex that moves processively toward the microtubule plus-end. (2) Although TRAK2 activates kinesin-1 in the absence of dynein/dynactin, the formation of a TRAK2/kinesin-1/dynein/dynactin complex further enhances the frequency of transport toward the plus-end. (3) TRAK2 binds dynein and dynactin to form a motor complex that moves processively toward the microtubule minus-end in the presence of LIS1. LIS1 transiently activates dynein before dissociating from the motor complex. (4) Kinesin-1 binding to TRAK2 further enhances TRAK2 motility toward the minus-end. (5) TRAK2 coordinates the activities of kinesin-1 and dynein–dynactin within the TRAK2/kinesin-1/dynein/dynactin complex to allow for processive transport in a single direction. **b** A model for mitochondrial transport by multi-motor TRAK2 complexes. The TRAK2/kinesin-1/dynein/dynactin complex associates with mitochondria via a Miro protein (gray). Kinesin and dynein are selectively activated or inhibited by cellular factors to promote mitochondrial transport toward the microtubule plus- or minus-end.

Smith, University of South Carolina), Myc-KIF5B (full-length mouse KIF5B in pRK5-Myc vector, gift from J. Kittler, University College London), Myc-KIF5C-Halo (full-length mouse KIF5C in pRK5-Myc vector, gift from J. Kittler, University College London, with a HaloTag subcloned into the C-terminus), KIF5C(1–560)-Halo (original KIF5C 1–560 construct, gift from Y. Konishi, Hamamatsu University, was subcloned into pEGFP-N1 plasmid backbone with eGFP removed), KIF5B(1–560)-Halo (original KIF5B 1–560 construct, gift from R. Vale, UCSF, was subcloned into the pHTC-HaloTag-CMV neo Vector from Promega), FLAG-p150^Glued (gift from T. Schwarz, Boston Children's Hospital), eGFP-P25 (gift from T. Schroer, Johns Hopkins University), and Halo-HOOK1(1–554) (first 554 aa of human HOOK1 were generated from the human HOOK1 sequence and subcloned into pFN21A-HaloTag-CMV vector from Promega). All constructs were verified by DNA sequencing.

For RNAi transfection in knockdown experiments, Lipofectamine RNAiMax (Invitrogen) was used for transfection of siRNA duplexes at 50 nM. The 5′ to 3′ sequence of each siRNA is as follows: control non-targeting—UGGUUUACAUG

UCGACUAA, KIF5B—GAACUGGCAUGAUAGAUGA, DHC—GAUCAAACA UGACGGAAUU, p150^Glued—CUGGAGCGCUGUAUCGUAA, TRAK1—GGAA ACGAUGAGCGGAGUA.

**Protein expression and purification**. KIF5C(1–560)-Halo and rigor kinesin-$1_{E236A}$ proteins were purified as described in Masucci et al.[65]. Briefly, the plasmids were transformed into BL21(DE3)pLysE bacteria (Sigma, CMC0015-20X40UL) and grown in Terrific Broth media at 37 °C then at 18 °C for 18 h in the presence of 0.15 mM IPTG. Cells were pelleted, flash frozen in liquid nitrogen, and stored at −80 °C. On the day of purification, cells were lysed by microfluidizer and clarified through centrifugation. The proteins were purified through a $Co^{2+}$ agarose bead column (GoldBio, H-310-25) according to the manufacturer's protocol. KIF5C (1–560)-Halo was then bound to newly polymerized microtubules with AMPPNP and pelleted via centrifugation. The bound motors were then released from microtubules with ATP and centrifuged to remove the microtubules.

**Immunoprecipitation assays**. COS-7 cells were harvested 20–24 h after transfection in 300 μL of lysis buffer containing 50 mM HEPES (pH 7.4), 25 mM NaCl, 1 mM EDTA, 1 mM MgCl₂, 0.5% Triton X-100, and protease inhibitors (1 mM PMSF, 0.01 mg/ml Nα-p-tosyl-L-arginine methyl ester, 0.01 mg/ml leupeptin, 0.001 mg/ml pepstatin A, 1 mM DTT). Cell lysates were clarified at $17,000 \times g$ for 10 min before use. Fifty microliters of Protein G Dynabeads (Promega) were resuspended in 200 μL of PBS + 0.02% Tween-20. The beads were incubated with 5 μg of anti-FLAG (Sigma, F4042), anti-GFP (Abcam, ab1218), or anti-HaloTag (Promega, G9281) for 15 min at room temperature. Beads were then washed once with 200 μL PBS + 0.02% Tween-20, once with 200 μL lysis buffer, and incubated with 300 μL lysate for 30 minutes at room temperature. Beads were then washed three times with 300 μL PBS, eluted in 60 μL SDS sample buffer, and analyzed by Western blot.

**Microtubule pelleting assays**. COS-7 cells were transfected with siRNAs against KIF5B, DHC, p150, or a control non-targeting siRNA for 40–48 h and transfected with Halo-TRAK2 for 20–24 h. Cells were then lysed in BRB80 buffer (80 mM PIPES, 1 mM EGTA, and 1 mM MgCl2 (pH 6.8)) with 0.5% Triton X-100 and protease inhibitors (as described above) and clarified with two centrifugation steps at 17,000 and $32,000 \times g$. An input sample was taken from the clarified cell lysate. Unlabeled tubulin was clarified at $352,000 \times g$ for 10 min at 4 °C to pellet potential tubulin aggregates then polymerized at 5 mg/ml in BRB80 and stabilized with 1 mM GMPCPP. Equal volumes of cell lysate were then incubated with 0 or 5 μM of GMPCPP-stabilized microtubules at 37 °C for 20 min. Samples were then centrifuged at $38,400 \times g$ at 25 °C for 20 min. The resulting supernatant and pellet were then separated and denatured. The denatured inputs and pellets were then analyzed by Western blot.

**Immunoblotting**. Samples were analyzed by SDS-PAGE and transferred onto PDVF Immobilon FL membranes (Millipore). Membranes were dried for 1 h, rehydrated in methanol, and stained for total protein (LI-COR REVERT Total Protein Stain). Following imaging of the total protein, membranes were destained, blocked for 1 h in Odyssey Blocking Buffer TBS (LI-COR), and incubated overnight at 4 °C with primary antibodies diluted in Blocking Buffer with 0.2% Tween-20. Membranes were washed four times for 5 min in 1xTBS Washing Solution (50 mM Tris-HCl pH 7.4, 274 mM NaCl, 9 mM KCl, 0.1% Tween-20), incubated in secondary antibodies diluted in Odyssey Blocking Buffer TBS (LI-COR) with 0.2% Tween-20% and 0.01% SDS for 1 h, and again washed four times for 5 min in the washing solution. Membranes were immediately imaged using an Odyssey CLx Infrared Imaging System (LI-COR). Band intensities were measured in the Licor Image Studio application. Primary antibodies used for Western blots included the following: HaloTag (Promega, G9281) at 1:1000, p150^Glued (BD Transduction Laboratories, 610474) at 1:5000, Kinesin Heavy Chain (Millipore, MAB 1614) at 1:1000, DHC (Santa Cruz Biotechnology, R-325) at 1:500, FLAG (Sigma, F4042) at 1:1000, Myc (Sigma, R950-25) at 1:1000, HA (Covance, 16B12) at 1:1000, LIS1 (Abcam, ab109630) at 1:1000, and TRAK1 (Thermo PA5-44180) at 1:1,000. Secondary antibodies used for Western blots included the following: IRDye 800CW donkey anti-rabbit IgG (LI-COR, 926-32213), IRDye 680RD donkey anti-rabbit IgG (LI-COR, 926-68073), and IRDye 800CW donkey anti-mouse IgG (LI-COR, 926-32212).

**Single-molecule motility assays**. COS-7 cells expressing Halo-tagged constructs for 20–24 h were labeled with 2.5 μM TMR-Halo ligand (Promega) for 15 min, washed twice with Ca²⁺- and Mg²⁺-free Dulbecco PBS (dPBS; GIBCO), returned to culture medium and left in the incubator for 30 min. For 3-color imaging experiments, GFP-DHC HeLa cells were labeled with TMR-Halo ligand and washed as described then labeled with 375 nM JF646-SNAP (provided by Luke Lavis, Janelia Farms) for 30 min, washed twice with dPBS, returned to culture medium and left in the incubator for 15 min. Cells were then washed twice with dPBS and collected in dPBS and pelleted at $5000 \times g$ for 5 min. The cell pellet was then lysed in 40 mM HEPES, 1 mM EDTA, 120 mM NaCl, 0.1% Triton X-100, and 1 mM magnesium ATP (pH 7.4) supplemented with protease inhibitors (as described above). The lysate was left on ice for 10 min and clarified at $17,000 \times g$ for 10 min at 4 °C.

GMPCPP-stabilized microtubule seeds were prepared by combining unlabeled tubulin and 2.5 or 5% HiLyte647-labeled or HiLyte488-labeled tubulin (Cytoskeleton) to a final concentration of 50 μM. This mix was clarified, polymerized, and stabilized as described above. A soluble tubulin mix was prepared by combining unlabeled tubulin and 2.5 or 5% labeled tubulin of the same color to a final concentration of 50 μM. This mix was clarified at $352,000 \times g$ for 10 min at 4 °C and kept on ice. In all experiments, the labeling of the microtubule seeds and free tubulin are the same.

Flow chambers were prepared using silanized #1.5 glass coverslips (Warner) attached to glass slides (FisherScientific) using double-sided tape. To reduce non-specific binding, coverslips were cleaned through rounds of sonication in acetone, potassium hydroxide, and ethanol, dried, plasma cleaned and silanized with PlusOne Repel-Silane (GE Healthcare). Flow chambers were coated with 0.5 μM rigor kinesin-1_E236A, washed and blocked with 5% Pluronic F-127 (Sigma). After washing, 25 nM-labeled GMPCPP-stabilized microtubules were flowed-in and left

to attach to the coverslip for 1 min. Unbound microtubules were washed out with P12 (12 mM PIPES, 1 mM EGTA, 2 mM MgCl2). A final solution with 1:20 cell lysate, 10 μM tubulin mix, 1 mM Mg-GTP, 1 mM Mg-ATP in Dynamic Assay Buffer (P12, 0.3 mg/mL BSA, 0.3 mg/mL casein, 10 mM DTT, 15 mg/mL glucose, 0.05% methylcellulose) and an oxygen scavenging system (0.5 mg/mL glucose oxidase, 470 U/mL catalase; Sigma) was then flowed-in and let to equilibrate for ~5 min at 37 °C before time lapse acquisition was initialized. Three to five videos lasting 3 min were acquired at 37 °C for each chamber. The microtubule channel was acquired at 1 frame each 10 s and the motor/adaptor channel was acquired at 4 frames per second in experiments examining a single motor or adaptor moving along dynamic microtubules. In dual-color co-migration experiments, the microtubule channel was acquired at 1 frame each 10 s and the other channels were acquired at 1 frame per second. In three-color co-migration experiments, all channels were imaged at 1 frame per second. Imaging was performed on a Nikon Eclipse Ti Inverted Microscope equipped with an Ultraview Vox spinning disk TIRF system and 100 × 1.49 NA oil immersion objective (Nikon). Signals were collected using a Hamamatsu EMCCD C9100-13 camera, with a pixel size of 158 nm, controlled by Volocity software (PerkinElmer).

**Analysis of motility on dynamic microtubules**. Kymographs of the Halo-TRAK2 or Myc-KIF5C-Halo channel were generated for each microtubule at its maximum length by plotting a segmented line and using the Multi Kymograph macro for ImageJ. Microtubules were randomly selected from the field of view, but only microtubules with clearly defined polarity were analyzed. The microtubule plus-end was identified as the end with higher rates of growth and catastrophe through the 3 min video. In 3-color co-migration experiments, the polarity of unlabeled microtubules was defined by the transport of KIF5C toward the plus-end. Individual runs were then scored as processive to the MT plus-end, processive to the MT minus-end, or bidirectional/diffusive. Each processive run was manually tracked to determine run length and velocity. Processive motility to either microtubule end was defined as unidirectional transport for a minimum of 4 pixels (632 nm) lasting at least 1 s (4 frames) without switching direction. In unidirectional trajectories that exhibited pauses of at least 1 s between processive segments, each segment was analyzed as a separate run. Bidirectional/diffusive motility was defined as any run lasting at least 2 s (8 frames) that traveled at least 4 pixels (632 nm) and switched direction during transport. Diffusive events were manually tracked to determine the net displacement from the position where the run initiated to the position where the run terminated. Run frequency was calculated on a per-video basis by dividing the total number of runs of each kind of motility by the total MT length. The number of runs per micron MT in each video was then divided by the TRAK2 or KIF5C concentration in each cell lysate. The relative concentration of TMR-labeled TRAK2 or KIF5C in each lysate was calculated by interpolating the fluorescence intensity value on a standard curve of TMR at known concentrations obtained from a Cytation 5 Imaging Reader (BioTek) using 554 ± 5 nm excitation band and 580 ± 5 nm emission band. TMR-normalized run frequencies were then averaged across videos to determine the run frequency per cell lysate. A final normalization was performed by scaling all data so the average frequency across biological replicates of the control condition is 1.

MSD analysis was performed by manually tracking the position of a single TRAK2 particle every frame during a run. The MSD was then calculated from the run trajectory in R version 4.0.3.

**Statistics and reproducibility**. Statistical analyses were performed using Graph-Pad Prism version 9.1.0. Two-tailed unpaired student's $t$ test, one-way ANOVA with Dunnett's multiple comparisons test, or two-tailed Mann–Whitney U test were used to calculate $p$ values. *$p < 0.05$, **$p < 0.01$, ***$p < 0.001$, and ****$p < 0.0001$ are considered significant. The types of the statistical tests, sample size, and statistical significance are reported in the figures and corresponding figure legends. Data in column graphs are shown as individual data points with mean ± standard deviation from biologically independent experimental replicates. Individual data points from technical replicates within experiments are plotted whenever possible. All statistical analysis is conducted on data from at least three biologically independent experimental replicates. Kymographs and western blot images shown in the figures are representative of three or more independent experiments with similar results unless otherwise noted. The source data for statistical analyses can be found in the Source data file.

**Reporting summary**. Further information on research design is available in the Nature Research Reporting Summary linked to this article.

## Data availability
Data supporting the findings of this study are available from the corresponding authors upon reasonable request. Source data are provided with this paper.

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

## Acknowledgements

We thank Mariko Tokito and Erin Masucci for technical assistance. We thank Elizabeth Gallagher, Stephen Coscia, and Sydney Cason for helpful feedback and discussion. This work was supported by National Institutes of Health Grants T32 GM008216 to A.R.F., AG064618 to T.A.J., and R35 GM126950 and RM1 GM136511 to E.L.F.H.

## Author contributions

A.R.F., T.A.J., and E.L.F.H. designed the experiments and wrote the manuscript. A.R.F. performed the experiments and analysis. All the authors reviewed the figures and manuscript and approved its final version.

## Competing interests

The authors declare no competing interests.
