## [Peer Review File · Nature Communications]

Reviewers' Comments:

Reviewer #1:

Remarks to the Author:

Fenton et al investigate TRAK2 as cargo adapter for dynein and kinesin-1. They perform primarily cell lysate-based motility assays and immunoprecipitations to show that TRAK2 moves towards plus and minus end or diffuses. The latter seems to be a property of TRAK2 itself similarly to what was shown for TRAK1 earlier. The directed motility depends on kinesin-1 and dynein based on RNAi treatment of lysates and in line with previous findings. In contrast to previous ideas that TRAK2 would primarily recruit dynein, the authors find that TRAK2 moves predominantly towards the plus end in extracts. Minus end-directed motility can be enhanced by overexpression of Lis1 in line with recent findings that Lis1 promotes efficient recruitment of dynein to cargo adapters. However, it is not clear whether the relatively poor performance of minus end-directed transport is due to the particular assay conditions that might favour kinesin-based motility or allow more kinesin to remain attached to TRAK2 during the cell lysate preparation. The key claim the authors make in the manuscript is that TRAK2, dynein, dynactin and kinesin-1 form a complex and that motility towards both plus and minus end does to some extent depend on the formation of that complex of those opposing motors. While this might be true, any direct evidence is missing from the manuscript. At no point can we see dynein, kinesin and TRAK2 moving together on the microtubule. The composition of TRAK2 particles studied is unclear and the depletion of specific motors from cells could change localisation of TRAK2, transcriptional profile of the cells, recruitment of other compensatory motors etc. Thus a range of alternative explanations is possible for the observations made.

I have also some issues with the quality of the data presented here:

Microtubule polarity is unclear at best in Fig. 1c. Actually, the supposed plus end shown does not change length in 4 seconds. If this was judged based on assembly speed of the ends, this should be shown in the figure. The motility analysis throughout the paper is based on the ability to determine polarity of microtubules accurately and thus data are required to convince the reader that this could be reliably determined for all microtubules analysed. If the example in S1 is the best they have, then I doubt the classification. S1a supposed minus end looks rather like the boundary to the seed – the methods don't reveal whether labelling between extensions and seed were labelled differently. The accuracy of determining run direction from microtubule dynamics should be verified by doing experiments with a single labelled motor – and reported in the manuscript. As we cannot be sure how reliably the microtubule polarity has been identified, we cannot be sure whether there even is motion of TRAK2 towards the minus end – the 5% of events shown in Figure 1 might as well have been wrongly classified microtubules.

There are some data that suggest some dynactin subunits to co-migrate with TRAK2 and interact with KIF5b in a TRAK dependent manner, but that dynein is part of such a complex is not shown. The data that show microtubule binding of TRAK2 depends on kinesin, dynein and dynactin don't look convincing. Have the authors taken into account that TRAK2 sedimented from control lysate in the absence of microtubules?

Data should be summarised not per video, but by biological repeat. The run frequency will depend a lot from the careful normalisation of TRAK labelling and total protein concentration of the extracts and hence comparing run frequencies across different lysates as it is done here needs to be repeated with different extracts and transfections rather than just taking a couple of movies. Otherwise any mistake in diluting the extract to comparable level would show as a significant phenotype. The normalisation procedure needs to be explained too. Why aren't absolute frequencies shown? This would also give an indication how reproducible the experiments are.

There seems to be a significant number of static events that have not been considered in the study. The frequency of those seems to increase in the CC1-Box mutants, so might be relevant to consider?

Reviewer #2:

Remarks to the Author:

The manuscript by Fenton et. al. provides mechanistic insight on how mitochondrial motor adaptor protein TRAK2 coordinates kinesin and dynein motors for mitochondrial transport. The authors investigate the role of TRAK2 by using in vitro reconstitution and single-molecule imaging approaches. Their biochemical and transport assays reveal that TRAK2 functionally links opposing motors by also acting as an activating adaptor for dynein. Interaction of both dynein and kinesin motors with TRAK2 (and TRAK1) has previously been demonstrated in several studies. However, whether the TRAK1/2 are just motor adaptors, or an activating adaptor to enhance the processivity of dynein-dynactin for mitochondrial transport, has been an open question in the field. Here, the authors provide evidence for the first time, by mutating highly conserved CC1-box motif, that TRAK2 may serve as a dynein activating adaptor, which is the main strength of this manuscript. This reviewer thinks that the manuscript by Fenton et.al. is appropriate for publication in Nature Communication after addressing the following comments:

- It is known that TRAK1 and TRAK2 can form heteromer. Mitochondrial motor adaptor protein Miro is also involved in dynein-dependent mitochondrial transport (López-Doménech et.al. 2018). Since COS7 cells have high level of TRAK1 and Miro proteins, it is expected that Halo-TRAK2 will have some level of interactions with these proteins, when purified from cell lysates. Further characterization of Halo-TRAK2 is necessary for accurate interpretation of the in vitro transport assays.
- TRAK2 displaying mostly plus-end motility may also be due to TRAK1's presence in the same complex. TRAK1 knock-down experiments would address these concerns.
- When TRAK2 is overexpressed, if endogenous Miro levels are not sufficient, it may only form complex with cytoplasmic motor proteins. Considering Miro's role in minus-end directed transport, Miro overexpression experiments will be very important. For example, instead of Lis1, Miro may be the processivity booster. Miro and Trak2/1 dependent coordination of kinesin and dynein motors for mitochondrial transport could be further investigated with truncated TRAK2 expression (TRAK2 lacking Miro binding domain) or Miro1/2 KO cells.
- In addition to siKIF5B, siDHC and sip150 (Figure 5), microtubule co-sedimentation assay should be performed with Halo-TRAK2 A/V.

Reviewer #3:

Remarks to the Author:

The authors investigate the role of TRAK2 in regulating bidirectional transport. They show that TRAK2 is a dynein activator, analogous to BicD or Hook, although it seems to need LIS1 to truly activate dynein. The LIS1 requirement tempers the "activator" label, but the mutagenesis of sequences shown to be important for other activators is good support of TRAK2 as an activator. This is an important contribution. They also show through a series of experiments that both kinesin and dynein bind to TRAK2. That is a second strong contribution. Finally, the analysis that the bidirectional events are actually TRAK2 diffusion is well done and this is an important point to make.

However, the data supporting the contention that dynein and kinesin-1 are interdependent when complexed with TRAK2 are weaker and I thought the text went beyond the data. The evidence for codependence is that inhibiting one motor reduces the landing rate of processive runs in the opposite direction. This is seen both for kinesin knockdown reducing minus-end run frequency and dynein complex knockdown reducing plus-end run frequency. But this is perhaps expected – if landing of the

complex on the microtubule involves activity of both motors, then inhibiting one would be expected to lower frequency of both plus- and minus-end runs. Formally, I guess you would have to say this is functional interdependence. But, importantly, the run length and velocity of the events in the opposite direction were not affected. I think this is an important point, and I think the data in Fig S3, particularly the run length should be put into the main text. The fact that there were no bidirectional single-molecule events that could clearly be attributed to the motors rather than simple diffusion is another point that argues that once on the microtubule, both motors do not coordinate their activities. The pull-down assays provide important information, but they also do not speak to the proportion of complexes that have both motors attached. The Fig 4b colocalization data is the key piece of data that supports the contention that both motors are on the complex simultaneously during processive movement, but the frequency of these events, the demonstration of minus-end movement of labeled kinesin motors, and other details of this activity are not presented. Thus, the authors should temper their language in promoting this idea of codependence. The fact that it is only landing and not motility that supports codependence is important and interesting and should be stated more clearly.

Related to this, Figure 6 was complicated and potentially misleading. As described above, the data in support are somewhat tenuous, and the +/- is a terrible way to show this – it implies quantitation that isn't there. Thus, the authors should rethink figure 6.

Overall, the data are a very important contribution to the literature, but the presentation should be tightened as described above.

Comments:

1. Page 6: "The variability in measurements of minus-end-directed velocity is due in part to the small number of minus-end runs observed under these conditions; only ~3% of TRAK2 motility was towards the minus-end whereas ~73% of motility was towards the plus-end (Fig. 1h)." – It is not clear how small numbers give more variability; the variance or SD should be independent of N. Please clarify.
2. Page 10: "Thus, the initiation of kinesin-dependent TRAK2 transport to the microtubule plus-end requires dynein and dynactin, independent of dynein activation via LIS1." This statement is not factual. The plus-end activity was reduced but is still there, it doesn't *require* them.
3. Fig 1d and f and Fig S3a. The exponentials fitting the 1-cdf curves appear to have an offset (sometimes negative). That should not be a free parameter in these fits – a 1-cdf curve must go to zero and the fit should similarly go to zero. These should be fixed throughout.
4. In Fig 1i, how was run displacement defined? Was it displacement before directional switch or the net displacement of the entire trace?
5. Is there a reason for using different buffers for the microtubule pelleting assays and the single-molecule motility assays (BRB80 vs P12)? It's possible that the relative frequency of plus-end, minus-end and diffusional motility could change strongly with different buffers. This point would be worth bringing up, as it would be unfortunate if the results depended strongly on using a very low ionic strength buffer. Also composition of P12 buffer is not described in methods.
6. What were the plus- and minus-end run frequencies of the Halo-TRAK2 and GFP-p25 co-complexes compared to that when COS-7 cells were only transfected with Halo-TRAK2? The authors showed convincing evidence that dynactin and kinesin-1 can simultaneously associate with TRAK2, however, it was still unclear whether the TRAK2 complex always contain dynein, dynactin, and kinesin-1. It remains possible that dynein and kinesin-1 compete for binding sites on the dynactin-TRAK2 complex (or dynein-dynactin compete with kinesin-1). A comparison of run frequencies in plus- and minus-end directions of Halo-TRAK2 vs TRAK2-dynactin co-complex may provide more insights. Also Page 9: "Moreover, we found TRAK2-dynactin complexes moving processively towards the microtubule plus-end." This is the strongest evidence that a tripartite complex is actually shown and hence is an important piece data, but the data are not shown. The authors should show the data supporting this statement.

We thank the reviewers for their thoughtful comments on our manuscript. We have addressed each point in the revisions as described below.

Reviewer #1

- Fenton et al investigate TRAK2 as cargo adapter for dynein and kinesin-1. They perform primarily cell lysate-based motility assays and immunoprecipitations to show that TRAK2 moves towards plus and minus end or diffuses. The latter seems to be a property of TRAK2 itself similarly to what was shown for TRAK1 earlier. The directed motility depends on kinesin-1 and dynein based on RNAi treatment of lysates and in line with previous findings. In contrast to previous ideas that TRAK2 would primarily recruit dynein, the authors find that TRAK2 moves predominantly towards the plus end in extracts. Minus end-directed motility can be enhanced by overexpression of Lis1 in line with recent findings that Lis1 promotes efficient recruitment of dynein to cargo adapters. However, it is not clear whether the relatively poor performance of minus end-directed transport is due to the particular assay conditions that might favour kinesin-based motility or allow more kinesin to remain attached to TRAK2 during the cell lysate preparation. The key claim the authors make in the manuscript is that TRAK2, dynein, dynactin and kinesin-1 form a complex and that motility towards both plus and minus end does to some extent depend on the formation of that complex of those opposing motors. While this might be true, any direct evidence is missing from the manuscript. At no point can we see dynein, kinesin and TRAK2 moving together on the microtubule.
 - We appreciate the close reading of our work. We also appreciate the point raised that we did not include direct evidence for a TRAK2-dynein-kinesin-1 complex. We prioritized this point in further experiments, and now include direct evidence of a co-complex moving together along microtubules and evidence of single TRAK2-motor complexes switching direction due to the coordinated activities of kinesin and dynein – please see Figure 5 of the revised manuscript.
- The composition of TRAK2 particles studied is unclear and the depletion of specific motors from cells could change localisation of TRAK2, transcriptional profile of the cells, recruitment of other compensatory motors etc. Thus a range of alternative explanations is possible for the observations made.
 - We are not sure that we fully understand the referee's point here. Since our work involves cell lysates, changes in TRAK2 localization or motor recruitment are not really relevant. It is indeed possible that depletion of specific motors could change the transcriptional profile of cells, but our biochemical evidence for motor interaction builds on previous genetic and cellular studies, so we do not feel that it is necessary to invoke a more complicated explanation.
- I have also some issues with the quality of the data presented here: Microtubule polarity is unclear at best in Fig. 1c. Actually, the supposed plus end shown does not change length in 4 seconds. If this was judged based on assembly speed of the ends, this should be shown in the figure. The motility analysis throughout the paper is based on the ability to determine polarity of microtubules accurately and thus data are required to convince the reader that this could be reliably determined for all microtubules analysed. If the example in S1 is the best they have, then I doubt the classification. S1a supposed minus end looks rather like the boundary to the seed – the methods don't reveal whether labelling between extensions and seed were labelled differently. The accuracy of determining run direction from microtubule dynamics should be verified by doing experiments with a single labelled motor – and reported in the manuscript. As we cannot be sure how reliably the microtubule polarity has been identified, we cannot be sure whether there even is motion of TRAK2 towards the minus end – the 5% of events shown in Figure 1 might as well have been wrongly classified microtubules.
 - We recognize the concerns that TRAK2 directionality is based on determining microtubule polarity reliably. We note that the basis for these concerns is likely that under the imaging conditions used here (1 frame/10 sec), the dynamics of the minus-

end are often hard to observe, but that the dynamics of the plus-end are always visible and obvious.

- Importantly, we have addressed the concerns raised in two ways. First, we have used purified KIF5B 1-560, lysate-based KIF5C 1-560, and lysate-based HOOK1 1-554 to demonstrate that tracking the dynamics of the growing microtubule ends over 3 minutes is a reliable method to distinguish the polarized transport of kinesin-1 and dynein (Supplementary Figure 1). These results demonstrate that kinesin-1 moves to identified microtubule plus-end with >98% consistency, while the well-characterized dynein activator HOOK1 moves toward the identified microtubule minus-end with >99% consistency. Second, we have included additional kymographs of TRAK2 moving along dynamic microtubules (Supplementary Figure 2). We now include a clear example of TRAK2 moving toward the microtubule minus-end without LIS1 added and an example of TRAK2 moving on a microtubule with visible minus-end growth and catastrophe. We believe that these additional examples provide sufficient evidence to support our classification of microtubule polarity.
- To further clarify the point about tubulin labeling, we now include the following sentence in our description of the single-molecule motility assays in the Methods section, “In all experiments, the labeling of the microtubule seeds and free tubulin are the same.”
- There are some data that suggest some dynactin subunits to co-migrate with TRAK2 and interact with KIF5b in a TRAK dependent manner, but that dynein is part of such a complex is not shown.
 - We appreciate this important point as we had no evidence showing that dynein forms a complex with TRAK2 and kinesin-1 in our initial submission. Importantly, we now provide direct evidence that dynein heavy chain forms a complex with KIF5C in a TRAK2-dependent manner and that KIF5C, TRAK2, and dynein co-migrate as a single complex along microtubules (Figure 5d-g).
- The data that show microtubule binding of TRAK2 depends on kinesin, dynein and dynactin don't look convincing. Have the authors taken into account that TRAK2 sedimented from control lysate in the absence of microtubules?
 - We appreciate this point, as a small amount of TRAK2 was observed pelleting in the absence of microtubules in the example shown. We now include a quantification of HA-TRAK2 pelleting in conditions without microtubules added (Fig. 4i). We observed a slightly higher amount of HA-TRAK2 pelleting from control lysates in the absence of microtubules in some experimental replicates, but not across all replicates. The differences between the control and knockdown conditions without microtubules are non-significant and minimal compared to the effect observed with microtubules added. As a result, we believe our data fully support our conclusion that kinesin-1, dynein, and dynactin promote TRAK2 binding to microtubules.
- Data should be summarised not per video, but by biological repeat. The run frequency will depend a lot from the careful normalisation of TRAK labelling and total protein concentration of the extracts and hence comparing run frequencies across different lysates as it is done here needs to be repeated with different extracts and transfections rather than just taking a couple of movies. Otherwise any mistake in diluting the extract to comparable level would show as a significant phenotype.
 - We agree that run frequency data should be summarized by biological replicate. We have updated all run frequency graphs to show frequencies per biological replicate. All statistical analysis of run frequency is now based on values from biological replicates. However, we believe that the run frequency values in each video are informative for the reader as an indication of variability between technical replicates.

As such, we have chosen to display run frequency data as a superplot containing both the biological replicate frequency and individual video frequency.

- The normalisation procedure needs to be explained too. Why aren't absolute frequencies shown? This would also give an indication how reproducible the experiments are.
 - We now include additional detail about how run frequencies were calculated and normalized in the Analysis of Motility on Dynamic Microtubules section of the Methods that describes each step of the normalization process.
 - We believe that normalization of the number of runs per micron microtubule to the concentration of TMR-labeled TRAK2 or KIF5C is the appropriate methodology for cell lysate-based single-molecule imaging. This normalization accounts for variability in transfection efficiency and TMR-labeling by scaling to the final amount of TMR-labeled protein present in the lysate used for single-molecule imaging.
- There seems to be a significant number of static events that have not been considered in the study. The frequency of those seems to increase in the CC1-Box mutants, so might be relevant to consider?
 - Thank you for the suggestion, but we typically observe a low level of non-specific binding of TRAK2 to the surface of the coverslip in our single-molecule assays. We are unable to distinguish this low level of static background from TRAK2-mediated binding to the microtubule. As a result, we did not find the analysis of these events to be informative for this study.

Reviewer #2

- The manuscript by Fenton et. al. provides mechanistic insight on how mitochondrial motor adaptor protein TRAK2 coordinates kinesin and dynein motors for mitochondrial transport. The authors investigate the role of TRAK2 by using in vitro reconstitution and single-molecule imaging approaches. Their biochemical and transport assays reveal that TRAK2 functionally links opposing motors by also acting as an activating adaptor for dynein. Interaction of both dynein and kinesin motors with TRAK2 (and TRAK1) has previously been demonstrated in several studies. However, whether the TRAK1/2 are just motor adaptors, or an activating adaptor to enhance the processivity of dynein-dynactin for mitochondrial transport, has been an open question in the field. Here, the authors provide evidence for the first time, by mutating highly conserved CC1-box motif, that TRAK2 may serve as a dynein activating adaptor, which is the main strength of this manuscript. This reviewer thinks that the manuscript by Fenton et.al. is appropriate for publication in Nature Communication after addressing the following comments:
 - We thank the referee for their positive comments and thoughtful suggestions, which are addressed below.
- It is known that TRAK1 and TRAK2 can form heteromer. Mitochondrial motor adaptor protein Miro is also involved in dynein-dependent mitochondrial transport (López-Doménech et.al. 2018). Since COS7 cells have high level of TRAK1 and Miro proteins, it is expected that Halo-TRAK2 will have some level of interactions with these proteins, when purified from cell lysates. Further characterization of Halo-TRAK2 is necessary for accurate interpretation of the in vitro transport assays.
 - This is an interesting point. We performed additional experiments to address the interactions of Halo-TRAK2 with TRAK1 and Miro proteins, as described below.
- TRAK2 displaying mostly plus-end motility may also be due to TRAK1's presence in the same complex. TRAK1 knock-down experiments would address these concerns.
 - This is an important point, which we addressed by performing siRNA knockdown of TRAK1 to assess possible effects on TRAK2 motility in the presence or absence of exogenous LIS1. As described in Supplementary Figure 6, we found no change in the frequency, run length, or velocity of TRAK2 transport toward either microtubule

end upon knockdown of TRAK1. While TRAK2 can associate with TRAK1, we find that the motility of TRAK2 in this *in vitro* system is independent of TRAK1.

- When TRAK2 is overexpressed, if endogenous Miro levels are not sufficient, it may only form complex with cytoplasmic motor proteins. Considering Miro's role in minus-end directed transport, Miro overexpression experiments will be very important. For example, instead of Lis1, Miro may be the processivity booster. Miro and Trak2/1 dependent coordination of kinesin and dynein motors for mitochondrial transport could be further investigated with truncated TRAK2 expression (TRAK2 lacking Miro binding domain) or Miro1/2 KO cells.
 - We recognized that Miro proteins play essential roles in the regulation of TRAK-based mitochondrial transport in cells. However, Miro1/2 are mitochondrial transmembrane proteins, which complicates direct analysis of their function in lysate-based assays examining soluble proteins. We examined the effect of Miro proteins on TRAK2 transport along microtubules in the presence of LIS1 by overexpressing Myc-tagged Miro1 or Miro2 (see below). We found that overexpression of Miro1 or Miro2 did not enhance TRAK2 motility, but instead impaired TRAK2 transport toward either microtubule end. We believe that membrane-associated Miro1/2 may impair TRAK2 transport by increasing the amount of TRAK2 that is associated with mitochondria and spun out of our lysate. Alternatively, soluble Miro proteins may inhibit TRAK2 function in cell lysates, as these proteins are meant to function together only when associated with a mitochondrial membrane. In either case, investigation of the functional interactions between TRAK2 and Miro1/2 will require additional experiments with approaches distinct from those described here, and are beyond the scope of the current study.

- In addition to siKIF5B, siDHC and sip150 (Figure 5), microtubule co-sedimentation assay should be performed with Halo-TRAK2 A/V.
 - We appreciate the suggestion to further characterize the TRAK2 CC1-Box mutants via microtubule pelleting assay. However, we have extensively examined these CC1-Box mutants using our more sensitive single-molecule assay and we were not sure what additional information might be gleaned from this particular experiment. Thus, we focused on providing thorough answers to the other points raised.

Reviewer #3

- The authors investigate the role of TRAK2 in regulating bidirectional transport. They show that TRAK2 is a dynein activator, analogous to BicD or Hook, although it seems to need LIS1 to truly activate dynein. The LIS1 requirement tempers the “activator” label, but the mutagenesis of sequences shown to be important for other activators is good support of TRAK2 as an activator. This is an important contribution. They also show through a series of experiments that both kinesin and dynein bind to TRAK2. That is a second strong contribution. Finally, the analysis that the bidirectional events are actually TRAK2 diffusion is well done and this is an important point to make.
- However, the data supporting the contention that dynein and kinesin-1 are interdependent when complexed with TRAK2 are weaker and I thought the text went beyond the data. The evidence for codependence is that inhibiting one motor reduces the landing rate of processive runs in the opposite direction. This is seen both for kinesin knockdown reducing minus-end run frequency and dynein complex knockdown reducing plus-end run frequency. But this is perhaps expected – if landing of the complex on the microtubule involves activity of both motors, then inhibiting one would be expected to lower frequency of both plus- and minus-end runs. Formally, I guess you would have to say this is functional interdependence. But, importantly, the run length and velocity of the events in the opposite direction were not affected. I think this is an important point, and I think the data in Fig S3, particularly the run length should be put into the main text. The fact that there were no bidirectional single-molecule events that could clearly be attributed to the motors rather than simple diffusion is another point that argues that once on the microtubule, both motors do not coordinate their activities. The pull-down assays provide important information, but they also do not speak to the proportion of complexes that have both motors attached. The Fig 4b colocalization data is the key piece of data that supports the contention that both motors are on the complex simultaneously during processive movement, but the frequency of these events, the demonstration of minus-end movement of labeled kinesin motors, and other details of this activity are not presented. Thus, the authors should temper their language in promoting this idea of codependence. The fact that it is only landing and not motility that supports codependence is important and interesting and should be stated more clearly.
 - We agree entirely with these points relating to the specific language describing the observed relationship between kinesin-1 and dynein-dynactin when in complex with TRAK2. We have tempered our language and use of “codependence” throughout the paper. We have replaced this language with direct descriptions of the frequency of transport initiation across experiments.
 - We also provide new evidence that kinesin-1 and dynein are simultaneously in complex with TRAK2 during processive movement (Fig. 5 and Supplementary Fig. 10), which we were able to obtain using a GFP-dynein cell line generously provided by Tony Hyman.

- Related to this, Figure 6 was complicated and potentially misleading. As described above, the data in support are somewhat tenuous, and the +/- is a terrible way to show this – it implies quantitation that isn't there. Thus, the authors should rethink figure 6.
 - We fully agree with this point and have removed the +/- representation of motor activity from Figure 6.
- Page 6: “The variability in measurements of minus-end-directed velocity is due in part to the small number of minus-end runs observed under these conditions; only ~3% of TRAK2 motility was towards the minus-end whereas ~73% of motility was towards the plus-end (Fig. 1h).” – It is not clear how small numbers give more variability; the variance or SD should be independent of N. Please clarify.
 - We agree and have removed this statement from the manuscript.
- Page 10: “Thus, the initiation of kinesin-dependent TRAK2 transport to the microtubule plus-end requires dynein and dynactin, independent of dynein activation via LIS1.” This statement is not factual. The plus-end activity was reduced but is still there, it doesn't *require* them.
 - We agree and have changed the wording of this sentence to “Thus, the initiation of kinesin-dependent TRAK2 transport to the microtubule plus-end is enhanced by dynein and dynactin, independent of dynein activation via LIS1.”
- Fig 1d and f and Fig S3a. The exponentials fitting the 1-cdf curves appear to have an offset (sometimes negative). That should not be a free parameter in these fits – a 1-cdf curve must go to zero and the fit should similarly go to zero. These should be fixed throughout.
 - We very much appreciate this close consideration of the accuracy of our exponential fits. We have now fixed all single-exponential fits throughout the paper.
- In Fig 1i, how was run displacement defined? Was it displacement before directional switch or the net displacement of the entire trace?
 - We have defined run displacement by adding the following sentence to the Analysis of Motility on Dynamic Microtubules section of the Methods: “Diffusive events were manually tracked to determine the net displacement from the position where the run initiated to the position where the run terminated.”
- Is there a reason for using different buffers for the microtubule pelleting assays and the single-molecule motility assays (BRB80 vs P12)? It's possible that the relative frequency of plus-end, minus-end and diffusional motility could change strongly with different buffers. This point would be worth bringing up, as it would be unfortunate if the results depended strongly on using a very low ionic strength buffer. Also composition of P12 buffer is not described in methods.
 - We apologize for not including the composition of P12 buffer and have now included this information in the methods. Our choice of buffers for microtubule-pelleting and single-molecule motility are based off of previously published protocols used in the lab (Olenick et al., 2016, J. Biol. Chem).
 - To test whether the ionic strength of our assay buffer was biasing our single-molecule results toward a certain kind of motility, we examined TRAK2 motility with LIS1 co-expressed in either P12 buffer (low ionic strength) or BRB80 buffer (higher ionic strength). We found that TRAK2 displayed a strong bias for plus-end-directed transport irrespective of the buffer used (see below). We conclude that the plus-end bias of TRAK2 in this system is not due to the ionic strength of the assay buffers tested here, and instead anticipate that this bias is determined by additional factors in the cell lysate or the relative binding affinities of the components of the multi-protein complexes examined here.

- What were the plus- and minus-end run frequencies of the Halo-TRAK2 and GFP-p25 co-complexes compared to that when COS-7 cells were only transfected with Halo-TRAK2? The authors showed convincing evidence that dynactin and kinesin-1 can simultaneously associate with TRAK2, however, it was still unclear whether the TRAK2 complex always contain dynein, dynactin, and kinesin-1. It remains possible that dynein and kinesin-1 compete for binding sites on the dynactin-TRAK2 complex (or dynein-dynactin compete with kinesin-1). A comparison of run frequencies in plus- and minus-end directions of Halo-TRAK2 vs TRAK2-dynactin co-complex may provide more insights. Also Page 9: “Moreover, we found TRAK2-dynactin complexes moving processively towards the microtubule plus-end.” This is the strongest evidence that a tripartite complex is actually shown and hence is an important piece data, but the data are not shown. The authors should show the data supporting this statement.
 - We appreciate this crucial comment regarding the formation of TRAK2-motor complexes containing kinesin-1 and dynein-dynactin. We have addressed this comment by providing new, direct evidence that dynein heavy chain forms a complex with KIF5C in a TRAK2-dependent manner and that KIF5C, TRAK2, and dynein co-migrate as a single complex (Figure 5d-g). We believe that the experiments presented in this updated manuscript more fully address the concerns related to the formation of TRAK2-motor complexes containing kinesin-1 and dynein-dynactin.
 - We have also included a quantification of the plus- and minus-end run frequencies of Halo-TRAK2 and GFP-p25 co-complexes as requested, as these data are also informative (Supplementary Fig. 9).

In summary, we have addressed each point raised in the three thoughtful critiques, and now include substantial new data that further strengthens our manuscript. We thank the editor and referees for helping us to improve our work, and hope that our revised study will now be found suitable for publication in *Nature Communications*.

Reviewers' Comments:

Reviewer #1:

Remarks to the Author:

The authors have added new data in response to my and other reviewer's concerns. The paper is much improved. However, I have a few issues related to the new data added.

Figure 5g and S10 shows three-colour TIRF data as requested and examples of co-migration of dynein, kinesin and TRAK2. As throughout the paper motility of TRAK2 was analysed, it would be good to provide the reader with some information on the proportion of TRAK2 events that are trak2 alone, kinesin+trak2, dynein+trak2 and kinesin+dynein+trak2 as this will allow interpreting the results. I have difficulties to reconcile the data in Figure 1i-j, which shows increased run frequency of kif5c in the presence of trak2 with the new data in S10, which shows that only a tiny fraction of KIF5C-TMR runs actually contain TRAK2. Likewise there are minus end-directed dynein runs, but none of them contain TRAK2. This somewhat weakens the evidence for TRAK2 as activator for kinesin-1 and dynein. I am also not convinced that we see any data supporting the notion of "integrated control of opposing motors" as stated in the abstract. Indeed the discussion compares the diffusive, motor-independent motion seen here for TRAK2 in extracts to the bidirectional motility of mitochondria in cells - which presumably depends on motors. Thus the authors actually show very nicely that we cannot assume from two similar emergent phenomena that the underlying mechanism is the same, but then fail to make this crystal clear in the discussion.

I also don't think that the experiment explains the motor co-dependence paradox. There is increased landing of TRAK2 complexes as a function of motors (although the dynein depletion data are not statistically significant!). This can be explained by the presence of more microtubule interaction domains alone as no run characteristics is changed. In the context of mitochondria transport, this rather large organelle can recruit more than one TRAK2 and therefore several motors that provide microtubule binding sites whether they are in a complex or not. Thus again, increased landing of single molecules of TRAK2 do not explain why you need opposite polarity motors in a single TRAK2 complex for mitochondrial transport.

Thus the authors need to rewrite the last two paragraphs of the discussion to make clear the limitations of the current experiments and maybe suggest future work that could probe these concepts further.

I would also suggest to remove Figure 6. It adds nothing to the paper. If the authors are keen to leave it in, it should show something new. The proposed complexes are already shown in 1b, 3a and 5f. The colour scheme isn't explained nor makes it easy to see the complexes, Lis1 doesn't appear in the complexes.

Run length data in 2d,g, 4c,f should show fit values as in 1e and 3d,f.

Once these minor changes are made, I would be happy to recommend publication in Nature Communications.

Reviewer #2:

Remarks to the Author:

In this revised manuscript, Fenton et.al. satisfactorily addressed a series of points raised by different reviewers. The authors added considerable amount of new data that significantly improved their work. I recommend the current version of the manuscript for publication in Nature Communications.

Reviewer #3:

Remarks to the Author:

The authors have fully responded to my comments. Nice work.

Response to the points raised by **Reviewer #1**:

- The authors have added new data in response to my and other reviewer's concerns. The paper is much improved. However, I have a few issues related to the new data added. Figure 5g and S10 shows three-colour TIRF data as requested and examples of co-migration of dynein, kinesin and TRAK2. As throughout the paper motility of TRAK2 was analysed, it would be good to provide the reader with some information on the proportion of TRAK2 events that are trak2 alone, kinesin+trak2, dynein+trak2 and kinesin+dynein+trak2 as this will allow interpreting the results. I have difficulties to reconcile the data in Figure 1i-j, which shows increased run frequency of kif5c in the presence of trak2 with the new data in S10, which shows that only a tiny fraction of KIF5C-TMR runs actually contain TRAK2. Likewise there are minus end-directed dynein runs, but none of them contain TRAK2. This somewhat weakens the evidence for TRAK2 as activator for kinesin-1 and dynein.
 - While we appreciate the recommendation, the requested additional quantifications are not possible given the approach we used in these studies. We employed a cellular expression system for the three-color TIRF experiment, which involves the overexpression and labeling of KIF5C and TRAK2 in cells that are stably expressing GFP-dynein heavy chain. In the resulting extracts, we do not know the labeling stoichiometry of KIF5C and TRAK2. The presence of both unlabeled and endogenous proteins in our extracts prevents us from accurately quantifying the proportion of TRAK2 events containing either motor, and also explains the observation of KIF5C and dynein runs without fluorescently-labeled TRAK2. We strongly disagree with the comment that these unlabeled runs weaken the evidence provided for the activation of kinesin-1 and dynein by TRAK2 given the clarity of the data showing that TRAK2 activates kinesin-1 and dynein (Fig. 1-3).
- I am also not convinced that we see any data supporting the notion of "integrated control of opposing motors" as stated in the abstract. Indeed the discussion compares the diffusive, motor-independent motion seen here for TRAK2 in extracts to the bidirectional motility of mitochondria in cells - which presumably depends on motors. Thus the authors actually show very nicely that we cannot assume from two similar emergent phenomena that the underlying mechanism is the same, but then fail to make this crystal clear in the discussion.
 - We appreciate the careful consideration of how our experimental findings relate to the bidirectional transport of mitochondria. However, we believe that our data fully support our claim of integrated control of opposing motors as we observed TRAK2-motor complexes quickly switch direction during processive movement. The immediate nature of these directional switches requires precise orchestration of the activities of kinesin and dynein within a single TRAK2-motor complex. Further, these immediate directional switches closely resemble mitochondrial transport in cells. Neuronal mitochondria are known to rapidly reverse their direction of transport during processive movement. Our model of TRAK2 function indicates that these immediate reversals result from modulating the activities of kinesin and dynein within TRAK-motor complexes. This kind of bidirectional mitochondrial transport is distinct from the short back and forth movements displayed by mitochondria, which we suggest are motor-independent. We have expanded and further clarified our discussion to distinguish between these distinct kinds of mitochondrial transport and relate our observations of TRAK2-motor complexes to each.
- I also don't think that the experiment explains the motor co-dependence paradox. There is increased landing of TRAK2 complexes as a function of motors (although the dynein depletion data are not

statistically significant!). This can be explained by the presence of more microtubule interaction domains alone as no run characteristics is changed. In the context of mitochondria transport, this rather large organelle can recruit more than one TRAK2 and therefore several motors that provide microtubule binding sites whether they are in a complex or not. Thus again, increased landing of single molecules of TRAK2 do not explain why you need opposite polarity motors in a single TRAK2 complex for mitochondrial transport. Thus the authors need to rewrite the last two paragraphs of the discussion to make clear the limitations of the current experiments and maybe suggest future work that could probe these concepts further.

- We do not agree with this comment. While the reviewer is correct that there will likely be multiple motor complexes across the surface of a motile mitochondrion that can simultaneously interact with a microtubule and further enhance productivity, our single-molecule data show this motor co-dependence at the level of single motor complexes. As a result, this interdependence must stem from molecular interactions within these TRAK-motor complexes. We recognize that the presence of additional microtubule-binding domains in the complex is just one possible explanation for this finding and have suggested alternate mechanisms in the discussion.
- I would also suggest to remove Figure 6. It adds nothing to the paper. If the authors are keen to leave it in, it should show something new. The proposed complexes are already shown in 1b, 3a and 5f. The colour scheme isn't explained nor makes it easy to see the complexes, Lis1 doesn't appear in the complexes.
 - We believe that Figure 6 is a key part of this manuscript as it integrates the findings of the paper into a cohesive working model. However, we appreciate the suggestions to improve our model by more explicitly including LIS1 in the complexes and emphasizing the organization of TRAK complexes on mitochondria, and have revised this figure accordingly.
- Run length data in 2d,g, 4c,f should show fit values as in 1e and 3d,f.
 - All run length data now includes tau values for single-exponential decay fits.
- Once these minor changes are made, I would be happy to recommend publication in Nature Communications.